# A family of photoswitchable NMDA receptors

**Shai Berlin[1†], Stephanie Szobota[1†‡], Andreas Reiner[1§], Elizabeth C Carroll[1], Michael A Kienzler[1], Alice Guyon[1,2], Tong Xiao[3], Dirk Trauner[4], Ehud Y Isacoff[1,5,6]***

[1]Department of Molecular and Cell Biology, University of California, Berkeley, Berkeley, United States; [2]Institut de Pharmacologie Moléculaire et Cellulaire, Université de Nice Sophia Antipolis, Nice, France; [3]Department of Chemistry, University of California, Berkeley, Berkeley, United States; [4]Department of Chemistry, Center of Integrated Protein Science, University of Munich, Munich, Germany; [5]Helen Wills Neuroscience Institute, University of California, Berkeley, Berkeley, United States; [6]Physical Bioscience Division, Lawrence Berkeley National Laboratory, Berkeley, United States

**\*For correspondence:** ehud@berkeley.edu

[†]These authors contributed equally to this work

**Present address:** [‡]Otonomy, Inc., San Diego, United States; [§]Faculty of Biology and Biotechnology, Ruhr University Bochum, Bochum, Germany

**Competing interests:** The authors declare that no competing interests exist.

**Abstract** NMDA receptors, which regulate synaptic strength and are implicated in learning and memory, consist of several subtypes with distinct subunit compositions and functional properties. To enable spatiotemporally defined, rapid and reproducible manipulation of function of specific subtypes, we engineered a set of photoswitchable GluN subunits ('LiGluNs'). Photo-agonism of GluN2A or GluN2B elicits an excitatory drive to hippocampal neurons that can be shaped in time to mimic synaptic activation. Photo-agonism of GluN2A at single dendritic spines evokes spine-specific calcium elevation and expansion, the morphological correlate of LTP. Photo-antagonism of GluN2A alone, or in combination with photo-antagonism of GluN1a, reversibly blocks excitatory synaptic currents, prevents the induction of long-term potentiation and prevents spine expansion. In addition, photo-antagonism in vivo disrupts synaptic pruning of developing retino-tectal projections in larval zebrafish. By providing precise and rapidly reversible optical control of NMDA receptor subtypes, LiGluNs should help unravel the contribution of specific NMDA receptors to synaptic transmission, integration and plasticity.

## Introduction

NMDA receptors are ligand-gated ion channels at excitatory synapses throughout the nervous system. They trigger long-term potentiation (LTP) and long-term depression (LTD) of synaptic strength and are implicated in memory formation, synapse development, circuit refinement, neuropsychiatric disorders, excitotoxicity and neurodegeneration (*Lau and Zukin, 2007*; *Nabavi et al., 2014*). NMDA receptors are heterotetramers, consisting of two glycine-binding GluN1 subunits paired with two glutamate-binding GluN2 and/or glycine-binding GluN3 subunits, yielding a combinatorial diversity of channel composition and function (*Paoletti et al., 2013*; *Traynelis et al., 2010*). The glutamate-binding GluN2 subunits (GluN2A-D) are encoded by four genes, which appear at specific developmental stages, bind distinct regulatory proteins, and are situated at diverse cellular locations (*Hardingham and Bading, 2010*). The complexity of subunit stoichiometry and receptor localization has made it difficult to unravel the roles of NMDA receptor signaling in circuit function and behavior.

Much has been learned about the function of NMDA receptors from pharmacological and genetic manipulations that target specific receptor subtypes (*Foster et al., 2010*; *Tang et al., 1999*; *Traynelis et al., 2010*). However, pharmacological agents need to be used at low concentrations to maintain selectivity, resulting in slow onset, and are slow to wash out due to high affinity,

**eLife digest** Within the nervous system, neurons are organized into extensive networks via connections called synapses. To signal across a synapse, one neuron releases chemical messengers that bind to "receptors" on the surface of the neighboring cell. The ease with which neurons can communicate can change depending on how often a synapse is used. This adaptability is known as synaptic plasticity, and is central to the formation of memories.

NMDA receptors are one group of receptors that play an important role in synaptic plasticity. There are several types of NMDA receptor, which are made up of different combinations of protein subunits and have different properties. This means that each type contributes to synaptic plasticity in a slightly different way.

Other receptors found in neurons have been studied using a technique called chemical optogenetics, which allows the activity of modified proteins to be turned on and off by light. Now, Berlin, Szobota et al. have designed a toolbox that enables the activity of four of the NMDA receptor subunits to be controlled with light, which activates or blocks the NMDA receptors that they form (which includes several of the main receptor types). Thus, how these types of NMDA receptor contribute to synaptic plasticity can be investigated.

The toolbox can be used to control synaptic plasticity under a wide range of conditions. Plasticity can be induced or prevented in either single synaptic connections or large regions containing many thousands of synapses. The approach works in individual neurons grown artificially in the laboratory, in brain slices and in the living brain. Furthermore, synaptic plasticity can be controlled precisely to affect single synaptic events (which occur in milliseconds) or it can be controlled over several days to study whether this affects how neurons develop.

The next steps will be to expand the toolset so that the activity of all the NMDA receptor subtypes can be controlled using light. Further studies could then incorporate the receptors into the brains of mammals to study how the receptors' activity affects a range of processes including memory formation and disease.

are difficult to confine spatially, and generally cannot be targeted to a specific cell. Genetic manipulations, like gene-knockout or RNA-interference, provide subunit-specificity, but are either for the life of the organism or, when conditional, slow to turn on and cannot typically be turned off. These chronic effects can lead to circuit changes and compensation (*Nakazawa et al., 2004*; *Rossi et al., 2015*). Moreover, the ability to confine these manipulations to specific brain regions and cell types is limited. The problems of spatial and temporal control have been addressed by the advent of caged versions of glutamate, NMDA and the pore blocker MK-801, which can be photo-uncaged in very small volumes at precise times (*Huang et al., 2005*; *Kohl et al., 2011*; *Matsuzaki et al., 2001*; *Palma-Cerda et al., 2012*; *Rodríguez-Moreno et al., 2011*). MK-801 can work from inside the cell and, therefore, can be loaded *via* the patch pipet, whereas glutamate and NMDA cannot and so cannot be targeted precisely to a specific cell; furthermore, none of these compounds are selective for receptor subtype. A recent development has been the subunit-specific control of an ion channel with a photo-reactive unnatural amino acid that enables photo-inactivation. So far, this methodology has been applied to a potassium channel (*Kang et al., 2013*), AMPA receptor (*Klippenstein et al., 2014*) and GluN2B-containing NMDA receptors (*Zhu et al., 2014*). However, photo-inactivation requires intense and prolonged irradiation with UV light and, importantly, is irreversible.

To overcome the above obstacles, we set out to endow individual GluN subunits with fast and reversible light-switching via the site-directed, on-cell attachment of a Photoswitched Tethered Ligand (PTL). We employed PTLs from the 'MAG' family (*Figure 1a*), which consist of Maleimide (for covalent attachment to a cysteine residue substituted onto the water exposed surface of the ligand binding domain of the GluN subunit), a photo-isomerizable Azobenzene linked to a Glutamate ligand (for synthesis see [*Volgraf et al., 2006*]). Illumination with near UV light (360–405 nm; violet light) isomerizes MAG into the bent *cis*-configuration, whereas illumination with blue-green light (460–560 nm; green light) isomerizes MAG to the *trans*-configuration (*Figure 1a*) (*Gorostiza and Isacoff, 2008*). By choosing geometrically favorable positions for introduction of the cysteine

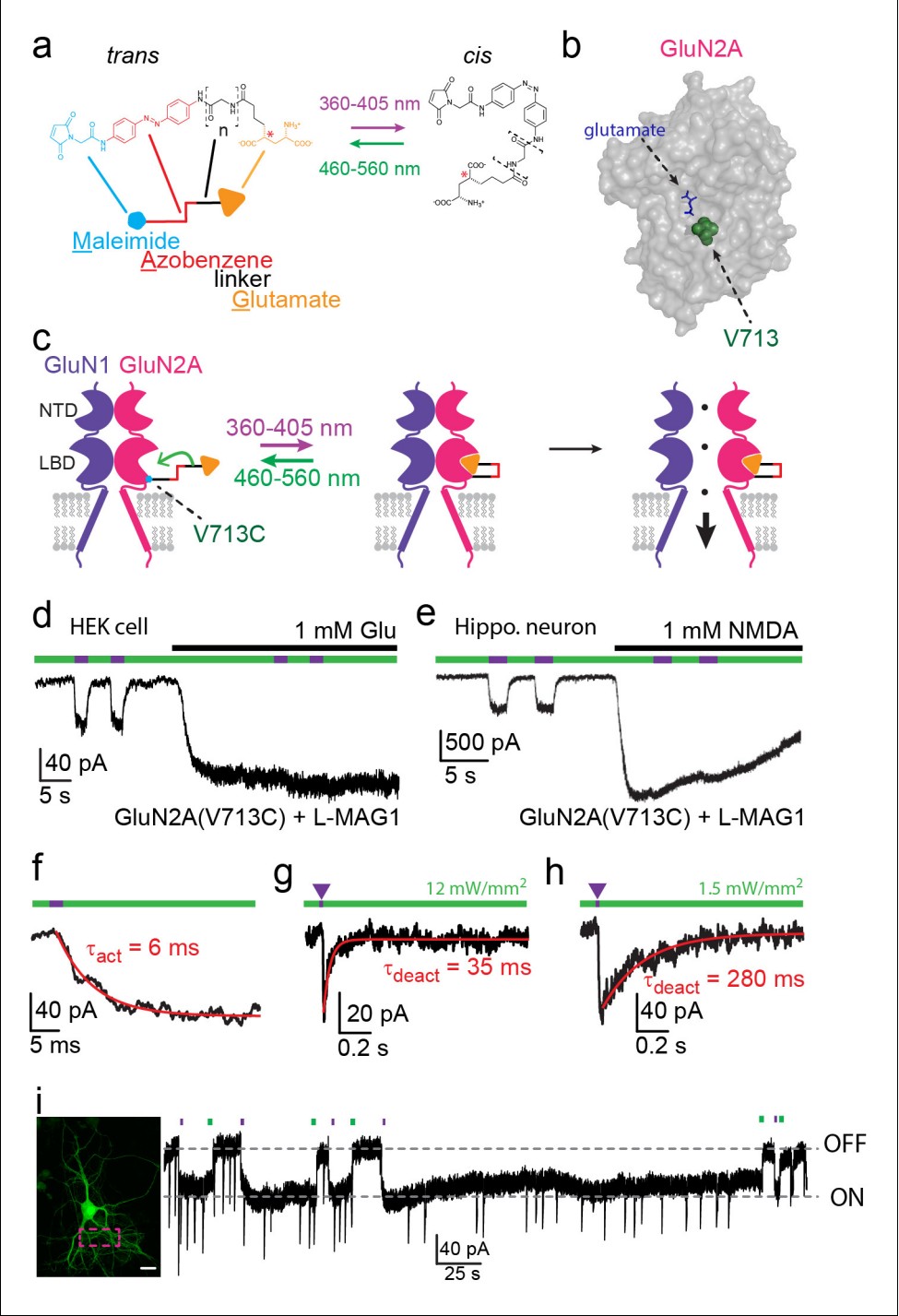

**Figure 1.** Photo-*agonism* of NMDA receptors in HEK293 cells and hippocampal neurons. (**a**) MAG photoswitches showing chemical structure and cartoon depiction. MAG0 and MAG1 differ in length (brackets, *n* = 0, 1) (*Gorostiza et al., 2007*), whereas L-MAG and D-MAG (*Levitz et al., 2013*) differ in stereochemistry (red asterisk). Illumination with 360–405 nm light photoisomerizes MAG from the elongated *trans*- azobenzene (red) configuration to the bent *cis* configuration; illumination at ~460–560 nm returns MAG to the *trans* isomer, as does slow thermal relaxation in the dark. The maleimide group (cyan) allows attachment to an engineered cysteine. (**b**) Space filled, crystal structure of GluN2A's Ligand Binding Domain (LBD, PDB-2A5S (*Furukawa et al., 2005*)) showing the site of glutamate binding (blue sticks) as well as the nearby V713 position (green spheres), which was mutated to a cysteine to which maleimide tethers and yields photo-agonism. (**c**) Cartoon depiction of a photo-agonized NMDA receptor showing, for simplicity, only one of the two LiGluN1a-*wt* subunits (purple) co-

*Figure 1 continued on next page*

*Figure 1 continued*

assembling with one of the two engineered LiGluN2A subunits (pink). The MAG photoswitch (color-coded sticks; as shown in a) is covalently attached to the LBD (cyan dot) to endow the channel with light-sensitivity. The *cis* configuration allows docking of the glutamate headgroup (orange triangle) into the binding pocket (middle cartoon), inducing LBD closure and channel opening (rightward cartoon). For simplicity the ligand of the GluN1 subunit (glycine) is not shown. Light- and glutamate/NMDA-induced currents in (d) HEK293 cells or (e) hippocampal neurons (from *wt* rats) expressing GluN1a-*wt* and GluN2A(V713C) labeled with L-MAG1. Photo-current is elicited by 380 nm light (~3 mW/mm$^2$) (violet bar) and turned off by 500 nm light (~3 mW/mm$^2$) (green bar). Full activation is induced by application of 1 mM glutamate or NMDA (black bar). (f) Representative trace (averaged from 4 consecutives sweeps) for fast MAG photoisomerization by an intense 375 nm light pulse (2 ms at ~20 W/mm$^2$) leading to rapid activation and opening of GluN2A(V713C) (red trace is a monoexponential fit, τ indicated). (g-h) Representative traces (2 cells) for tuning the off kinetics of the photo-current by applying high intensity (~12 mW/mm$^2$, average of 8 consecutive sweeps) for fast deactivation (g) or low green light intensity (~1.5 mW/mm$^2$, average trace from 4 consecutives sweeps) for slow deactivation (h). (i) Representative image (left, scale bar 10 µm) and trace (right) showing the bistability of the photoswitch in a cultured hippocampal neurons (from *wt* rats) transfected with GluN2A(V713C)-only. The *cis* state of MAG is photo-stable, so that following illumination (targeted line-scanning) by a brief 405 nm laser pulse (2 s) over a large region of the dendritic tree (dashed violet box) induces an inward current (black trace, ON) that persists in the dark without visible decay for tens of seconds (see also *Figure 1—figure supplement 3*). Likewise, following closure of the channel (black trace- OFF) with brief green light (488 nm) illumination (2 sec), the channel remains shut and no current is observed unless triggered anew by violet light. Spontaneous EPSCs are observed during the opening and closing of the channel.

The following figure supplements are available for figure 1:

**Figure supplement 1.** Screen of GluN2A cysteine positions and MAG variants.

**Figure supplement 2.** Pharmacological characterization of GluN2A(V713C), GluN2A(G712C), GluN2B(V714C) and GluN1a(E406C).

**Figure supplement 3.** Lack of perturbation of neurons by MAG or LiGluN2A(G712C).

---

attachment site (*Figure 1b* and *Figure 1—figure supplement 1*) and altering the length of the MAG molecule by varying the linker via additional glycine(s) (*Figure 1a*, dashed brackets; *n*), the glutamate moiety of MAG can be designed to either engage or obstruct the ligand binding pocket in the *cis* configuration, and withdraw from the ligand binding pocket in the *trans* configuration, yielding light-dependent gating (*Figure 1c* and *Figure 2a*, but see also [*Numano et al., 2009*]). PTLs, including MAGs, have been employed to generate light-gated ionotropic kainate receptors (*Janovjak et al., 2010*; *Reiner et al., 2015*; *Szobota et al., 2007*; *Volgraf et al., 2006*), metabotropic glutamate receptors (*Levitz et al., 2013*), nicotinic acetylcholine receptors (*Tochitsky et al., 2012*), P2X receptors (*Lemoine et al., 2013*) and GABA$_A$ receptors (*Lin et al., 2014*).

We now report a novel family of four <u>Li</u>ght-gated <u>GluN</u> subunits, or LiGluNs: 1) a light-activated GluN2A, 2) a light-activated GluN2B, 3) a light-antagonized GluN2A and 4) a light-antagonized GluN1, isoform 1a, (GluN1a). The first three LiGluN subunits enable selective manipulation of GluN2A- or GluN2B-containing receptors, whereas the fourth operates as a general controller of <u>all</u> plasma membrane NMDA receptors, owing to the obligatory occurrence of GluN1 in all NMDA receptors. We show that LiGluN-containing NMDA receptors function normally, incorporate into synapses, and that their expression does not alter NMDA receptor expression levels. Photo-switching can be sculpted to generate NMDA receptor currents that mimic the fast (GluN2A-like) or slow (GluN2B-like) deactivation kinetics of native excitatory postsynaptic currents (EPSCs). Widefield illumination and photo-activation of LiGluN2A or LiGluN2B-containing receptors in primary hippocampal neurons robustly drives activity, whereas photo-antagonism of LiGluN2A and LiGluN1a reversibly block excitatory synaptic currents. Spatially-targeted photo-activation of LiGluN2A-containing receptors on single dendritic spines can be used to trigger a spine-specific increase in calcium and the spine expansion that is associated with LTP. Complementarily to this, photo-antagonism of LiGluN2A-containing receptors can block LTP-induction via Schaffer collateral stimulation or prevent spine expansion. LiGluNs fulfill a major promise of chemical optogenetic photo-pharmacology by providing the kind of spatio-temporally precise control that is obtained with

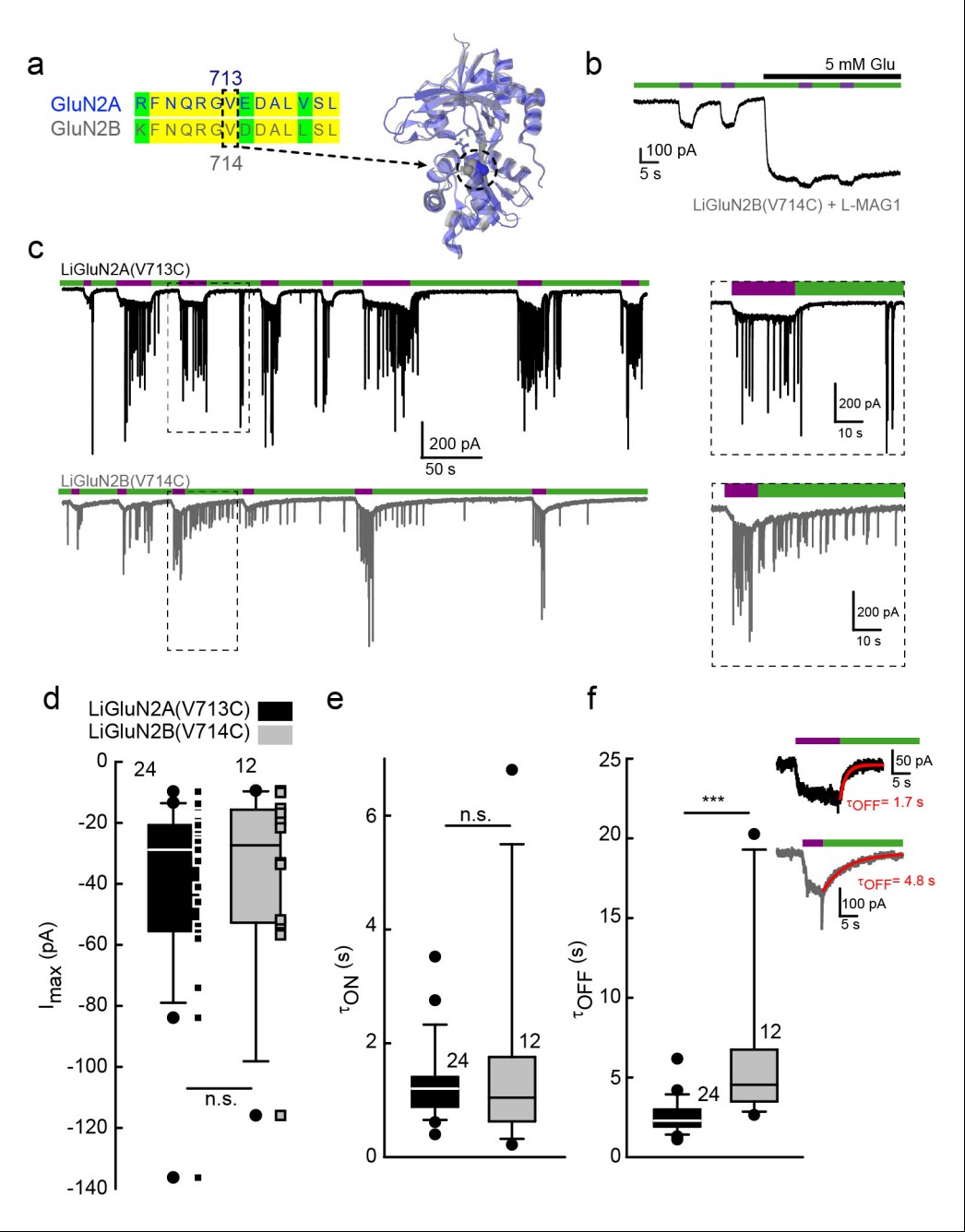

**Figure 2.** Rapid development of light-agonized LiGluN2B subunit, based on LiGluN2A. (**a**) Partial sequence alignment (left) and overlaid crystal structures (ribbon) of the LBDs of GluN2A (blue, PDB-2A5S (*Furukawa et al., 2005*)) and GluN2B (grey, PDB- 4PE5 [*Karakas and Furukawa, 2014*]) showing the high degree of similarity between the two LBDs and the corresponding mutation in GluN2B(V714) to that of LiGluN2A(V713) (dashed circle, Valine; color-coded spheres). (**b**) Representative trace of photo-agonism of LiGluN2B(V714C). A HEK293 cell transfected with GluN1a and LiGluN2B(V714C), and labeled with L-MAG1, when illuminated with 380 nm light (violet bars) produces an inward photo-current that can be turned off by 510 nm light (green bars). Note the small increase in current during glutamate perfusion, suggesting that L-MAG1 may act as a stronger agonist than glutamate. (**c**) LiGluN2B(V714C) photocurrents are of similar size, but slower to turn off than those of LiGluN2A (V713C). Hippocampal neurons (from *wt* rats), transfected with either LiGluN2A (top black trace) or -2B (bottom grey trace) exhibit similar inward photo-currents, during which barrages of spontaneous EPSCs emerge (violet bars), demonstrating that photo-activation of LiGluNs receptors causes action potential firing of presynaptic neurons (see also *Figure 4* and Figure 4 supplements). LiGluN2B photocurrents deactivate ~3 times slower than

*Figure 2 continued on next page*

*Figure 2 continued*

those of LiGluN2A (insets), as well as display consistently longer enduring observable EPSCs, summarized in **d-f**. (d) Box plot representation of median, outliers (filled circles) and individual data points (filled squares) are displayed. Red trace in (f) is a monoexponential fit, τ indicated. Statistics in panels are shown as Box plots of the data (not normally distributed, see *Figure 2—figure supplement 2*), showing median, outliers (filled circles) and individual data points (filled squares in **d**). Significance was tested using a nonparametric Mann-Whitney Rank Sum Test (see Materials and methods), ***p<0.001, n.s.- not-significant, *n* shown next to plots.

The following figure supplements are available for figure 2:

**Figure supplement 1.** Photo-activation of LiGluN2A and -2B drives action potential firing in hippocampal neurons.

**Figure supplement 2.** Summary of nonparametric statistics for *Figure 2*.

---

heterologous microbial opsins, that over-ride normal cellular signals, but in this case to control the native neuronal signaling proteins of the synapse that play central roles in synaptic plasticity, learning and memory. Together, these tools may open the door to advanced studies of receptor biophysics, and their function in synapses and neural circuits.

## Results

### Development of photo-agonized LiGluN subunits

We, and others, have previously generated light-controlled receptors and channels to manipulate cellular excitability (e.g. see reviews [*Kramer et al., 2013*; *Szobota and Isacoff, 2010*; *Tye and Deisseroth, 2012*]) by photo-regulating the flux of ions across the cell membrane. Here, we asked whether light-activated NMDA receptors could be engineered, which would traffic to synapses and function normally and thereby engage the cellular mechanisms of neuronal plasticity. Our previous success with light-gated glutamate receptors (*Levitz et al., 2013*; *Szobota et al., 2007*) along with other reports demonstrating that exogenously expressed GluN2 subunits efficiently co-assemble with endogenous GluN1 subunits and traffic properly to synapses (*Barria and Malinow, 2002*), suggested to us that this could be achieved by designing GluN subunits that contain a single cysteine attachment site for anchoring the MAG PTL.

We therefore systematically introduced single cysteine point mutations at different locations on the GluN2A ligand-binding domain (LBD) with proximity and accessibility to the glutamate binding site (*Figure 1—figure supplement 1a*), and following conjugation to different MAG variants (*Figure 1a*), assessed the effects of photoswitching in HEK293 cells, first using calcium-imaging and then following with voltage-clamp recordings of the most promising candidates, as shown below. We found several attachment positions that yielded light-responses, however focused on variants that when conjugated to L-MAG0 or L-MAG1, yielded the largest light-responses.

We initially focused on photo-agonism of the light-gated GluN2A subunits with the cysteine attachment site introduced at residue 713 (LiGluN2A(V713C)). When LiGluN2A(V713C) was co-expressed with wildtype (*wt*) GluN1a, photoisomerization of L-MAG1 to its *cis* configuration by illumination with 380 nm light (*Figure 1a,c*) generated an inward current (*Figure 1d,e*- violet bars) in both HEK293 cells (37.3 ± 2.2% of the total current induced by 1 mM glutamate, n = 5, *Figure 1d*) and in primary cultured hippocampal neurons (30.8 ± 5.9% of the total current induced by 1 mM NMDA, n = 6, *Figure 1e*). This photocurrent could then be completely turned off by photoisomerization of L-MAG to *trans* by illumination at 488 nm (*Figure 1d,e*- green bars). Since the isomerization of the azobenzene is fast (*Figure 1f*) (*Reiner and Isacoff, 2014*) and fully reversible, opening and closing of the channels could be accomplished by toggling between near-UV and green-light illumination, at various frequencies and repeatedly, to mimic various physiological patterns (*Figure 1—figure supplement 1b,c*). No inhibition of the current was observed during photo-agonism in the presence of saturating glutamate or NMDA, suggesting that when MAG in its *cis* form competes with free glutamate in LiGluN2A, and is as good an agonist as glutamate.

Dose-response profiling of NMDA receptors containing LiGluN2A(V713C) (as well as other variants, see below) shows preservation of the glutamate potency (EC50) as seen in *wt* receptors that contain the GluN2A-*wt* subunit (*Figure 1—figure supplement 2a,b*) (*Anson et al., 1998*; *Chen et al., 2005*; *Hedegaard et al., 2012*; *Paoletti et al., 2013*). Moreover, LiGluN2A(V713C)-containing receptors have *wt* activation kinetics, as assessed by fast MNI-glutamate uncaging that was designed to mimic the brief rise of glutamate that occurs in synapses due to vesicle release (*Figure 1—figure supplement 2g*).

Importantly, MAG labeling had no effect on the health, resting membrane potential or membrane resistance of cultured hippocampal neurons and yielded no photo-responses in neurons that did not express the cysteine-modified GluN2 subunit (*Figure 1—figure supplement 3a–c*), indicating a lack of action on native glutamate receptors or other channels. Moreover, expression of the LiGluN2A subunit did not significantly change the amplitude of NMDA-induced current in hippocampal neurons (*Figure 1—figure supplement 3d*), suggesting that it competes with native LiGluN2 subunits for assembly with the native GluN1, so that the total number of synaptic NMDA receptors is preserved, but now subject to photo-control. Together, these results show that NMDA receptors containing the LiGluN subunits should support normal synaptic transmission and plasticity, while providing specific photo-control over these processes.

## Temporal control of LiGluN2A(V713C)-containing receptors

MAG photoswitching provides two advantageous properties for the remote control of NMDA receptors. First, since the speed of MAG photoswitching depends on light intensity so that switching can be achieved either with short high intensity pulses or longer low intensity pulses (*Gorostiza et al., 2007*; *Reiner and Isacoff, 2014*), it should be possible to sculpt the photocurrent to resemble NMDA receptor excitatory postsynaptic currents (EPSC$_{NMDA}$). Using LiGluN2A(V713C) conjugated to L-MAG1, a 2 ms pulse of 375 nm light could be used to trigger substantial receptor activation (*Figure 1f*), with the fast rise kinetics comparable to those of EPSC$_{NMDA}$ kinetics observed for GluN2A-containing synaptic NMDA receptors (*Bidoret et al., 2009*; *Erreger and Traynelis, 2005*; *Vicini et al., 2001*; *Yuan et al., 2009*). Then, off-photoswiching could be triggered over tens of milliseconds (*Figure 1g*), as typical of GluN2A, or more slowly (*Figure 1h*) to mimic the slower GluN2B deactivation kinetics (*Erreger and Traynelis, 2005*; *Vicini et al., 1998*). Triggering these EPSC$_{NMDA}$ waveforms by only regulating light intensity avoids the need for piezo-driven fast perfusion systems and a cocktail of inhibitors typically used to isolate EPSCs$_{NMDA}$. Since variations in the off-kinetics leads to variation in the intracellular Ca$^{2+}$-concentration, this kind of control could be employed to study the role of Ca$^{2+}$ and specific NMDA receptor subtypes in the induction of LTP and LTD, as it is still debated how the magnitude, temporal pattern and NMDA receptor subunit type contribute to plasticity changes (*Cummings et al., 1996*; *Köhr et al., 2003*; *Pawlak et al., 2005*; *Shipton and Paulsen, 2014*; *Zhou et al., 2005*). A second advantageous property is the bistability of MAG photoswitches (*Gorostiza et al., 2007*). After switching with a brief 380 nm light pulse, photo-agonism (or photo-antagonism, as described below) is sustained in the dark for extended periods of time, until it is reversed by a brief 500 nm light pulse, as illustrated in both hippocampal neurons, (*Figure 1i*) and HEK293 cells (*Figure 1—figure supplement 3e*).

## Design of a photo-agonistic LiGluN2B subunit

Our scan to identify effective cysteine attachment sites for MAGs in LiGluN2A provided a guide for the development of a LiGluN2B based on the homology of its LBD with that of GluN2A. Hence, we introduced a cysteine at position 714 of GluN2B, corresponding to the 713 position that yielded the photo-agonism in GluN2A (*Figure 2a*). Indeed, conjugation of L-MAG1 to LiGluN2B (V714C) that was coexpressed with GluN1a in HEK293 cells yielded GluN2B-containing NMDA receptors which were activated by light (*Figure 2b*). Unlike with LiGluN2A(V713C), photo-activation of LiGluN2B(V714C) in the presence of saturating glutamate yielded a small increase in current (*Figure 2b*), suggesting that L-MAG1 is a more potent agonist than glutamate. Importantly, LiGluN2B(V714C)-containing receptors displayed the same glutamate affinity and activation kinetics as *wt* GluN2B receptors (*Figure 1—figure supplement 2c, d and h*), indicating that, as with LiGluN2A(V713C), the orthogonal light control is obtained over receptors that otherwise function normally.

Light-agonistic LiGluN2A and LiGluN2B subunits were examined side-by-side in hippocampal neurons. We expressed either LiGluN2A or LiGluN2B alone, relying on them to co-assemble with the endogenous pool of GluN1a subunits (*Barria and Malinow, 2002*). Following labeling with L-MAG1, each of these variants gave rise to photocurrents in neurons (*Figure 2c*). The LiGluN2A and LiGluN2B photocurrents were similar in amplitude and activation kinetics (*Figure 2d and e* and *Figure 2—figure supplement 2*), but the LiGluN2B photocurrent turned off ~3 times more slowly under identical light conditions (*Figure 2c*, inset and f and *Figure 2—figure supplement 2*), reminiscent of the slower deactivation kinetics of GluN2B- than GluN2A-containing receptors (*Bidoret et al., 2009*; *Vicini et al., 1998*). Widefield photoactivation at 380 nm of both LiGluN2A and LiGluN2B triggered an increase in EPSC frequency and this was reversed—more slowly in LiGluN2B—by illumination at 510 nm (*Figure 2c*), suggesting that photoactivation of NMDA receptors containing either LiGluN2A or LiGluN2B triggered action potential firing in presynaptic neurons. Indeed, in current clamp recording, photoactivation (violet bars) and photodeactivation (green bars) reliably and reproducibly elicited bouts of firing (*Figure 2—figure supplement 1a–e*). Photoactivation of LiGluN2A elicited a faster rise of excitation (*Figure 2—figure supplement 1a–b*, top insets, and f) and photodeactivation of LiGluN2A elicited a faster (~three fold) de-excitation (*Figure 2—figure supplement 1a–b*, bottom insets, and g) than neurons expressing LiGluN2B.

## Development of photo-antagonized LiGluN subunits

Our initial screen also identified several LBD anchoring positions where light blocked the glutamate-induced current (*Figure 1—figure supplement 1a*), suggesting that, instead of docking correctly in the glutamate binding pocket, the ligand end of MAG could obstruct access of free glutamate to the binding pocket (*Figure 3a*). The largest photo-antagonism was generated by L-MAG0 anchored at G712C of the GluN2A subunit when expressed in hippocampal neurons ($56.4 \pm 2.9\%$ inhibition of the total current induced by 1 mM NMDA, n = 7) (*Figure 3b, c and f*), similar to the block observed in HEK293 cells ($65.8 \pm 4.3\%$ inhibition of the total current induced by 1 mM glutamate in HEK cells, n = 9). This photo-antagonism is on par with the antagonism of GluN2A by NVP-AAM077, when applied at concentrations that maintain subunit selectivity (*Monaghan et al., 2012*; *Neyton and Paoletti, 2006*; *Paoletti and Neyton, 2007*), and so, although incomplete (likely due to labeling efficiency, see below and *Figure 3—figure supplement 1*), would be predicted to have utility. Whereas the specific photo-antagonism of GluN2A enables fast, reversible inhibition of GluN2A-containing NMDA receptors *only*, we also sought to develop a general tool for photo-antagonizing *all* NMDA receptor-subtypes within a given neuron, by creating a photo-antagonized GluN1. The GluN1 subunit is activated by glycine or D-serine, but not glutamate (*Figure 3b*). In fact, GluN1 antagonists resemble MAG in that they consist of an α-amino acid attached to a larger structure that prevents closure of the LBD (*Inanobe et al., 2005*). We therefore reasoned that MAG tethered near the GluN1 ligand binding pocket would prevent normal agonist binding when photo-docked and would thereby function as a photo-antagonist. Indeed, a screen of cysteine mutants around the glycine-binding pocket identified a labeling site, E406C, where L-MAG0 functioned as photo-antagonist ($30.7 \pm 1.7\%$ inhibition of the total current neurons, n = 6) (*Figure 3b, d and f*), without affecting the glycine potency of the subunit (*Figure 1—figure supplement 2e,f*). When the photo-antagonizable GluN1a(E406C) and GluN2A(G712C) subunits were co-expressed in dissociated hippocampal neurons and incubated with L-MAG0, their combined effect resulted in $73.8 \pm 5.2\%$ (n = 4) photo-inhibition of the total NMDA-induced current in neurons (*Figure 3e,f*).

Cultured hippocampal neurons from *wt* rats (C57BL) transfected with the photo-antagonizing GluN2A(G712C) subunit in combination with photo-antagonizing GluN1a(E406C), labeled with L-MAG0 and under green light illumination, exhibited typical action potential activity under current clamp recordings. Photo-antagonism with 370 nm light hyperpolarized cells and suppressed firing (*Figure 4a,b*), as seen with conventional NMDA receptor blockers. No such photo-effect was seen in non-transfected cells (*Figure 4c,d* and *Figure 4—figure supplement 2*). Reversal of photo-antagonism by 510 nm light elicited a gradual repolarization, reminiscent of physiological recovery (*Figure 4b*). In voltage clamp, neurons transfected with the photo-antagonizing GluN2A(G712C) subunit alone, or in combination with the photo-antagonizing GluN1a(E406C), displayed typical barrages of spontaneous $EPSC_{NMDA}$ events (*Groc et al., 2002*; *Ivenshitz and Segal, 2010*; *Kirson and Yaari, 1996*) (*Figure 4—figure supplement 1a*). During green light illumination, action currents were detected in some cells (*Figure 4—figure supplement 1a*, arrow and inset) and illumination

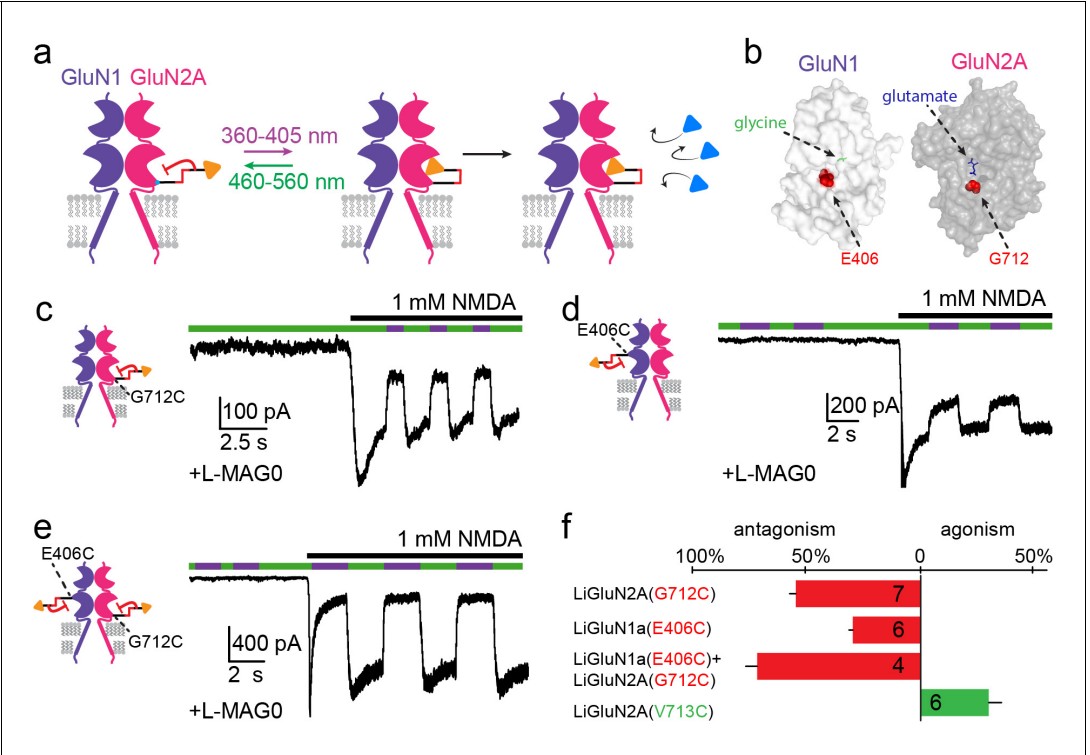

**Figure 3.** Photo-*antagonism* of NMDA receptors in hippocampal neurons. (**a**) Photo-antagonism with L-MAG0 attached to LiGluN2A(G712C), where the *cis*-configuration is thought to place the glutamate end of MAG near the binding pocket, where it impedes LBD closure or entry of free glutamate (blue triangles, rightmost cartoon), leaving the channel closed. (**b**) Space filled, crystal structures of GluN1a (light grey) and GluN2A (dark grey) LBDs (PDB-2A5S, 2A5T(*Furukawa et al., 2005*), respectively) showing the sites of glycine and glutamate binding (green and blue sticks, respectively) as well as the nearby E406 and G712 positions (red spheres) which, when tethered to L-MAG0, yield photo-antagonism. (**c-e**) Representative traces of photo-antagonism by 380 nm light (violet bars) of NMDA currents induced by 1 mM NMDA (black bars) in neurons (from *wt* rats) labeled with L-MAG0 that express either: **b**- GluN1a-*wt* and GluN2A(G712C) to photo-block the glutamate binding pocket of the GluN2A subunit; **c**- GluN1a(E406C) to photo-block the glycine binding site of GluN1a subunit; **d**- GluN1a(E406C) and GluN2A(G712C) to photo-block both classes of binding sites. (**f**) Summary of the average (± SEM) photo-antagonism (red bars) and photo-agonism (green bars) observed in neurons. *n* shown within bars.

The following figure supplement is available for figure 3:

**Figure supplement 1.** Moderate correlation between total current size and photo-current.

with violet light decreased the inward current, the frequency of action currents (*Figure 4—figure supplement 1a*, inset), and of spontaneous EPSC<sub>NMDA</sub> (*Figure 4—figure supplement 1a*, and summary in d), with no effect of violet light on non-transfected neurons (*Figure 4—figure supplement 1b*, right panel, summary in c and *Figure 4—figure supplement 2*).

## LiGluNs operate at synapses to provide optical control over NMDA receptor synaptic current

The fundamental power of the highly selective photo-pharmacology that is utilized in chemical opto-genetics is the ability to orthogonally and spatially control native signaling proteins in their physiological location in the cell- in the synapse for NMDA receptors. To test whether LiGluN2 subunits are incorporated into synapses, as shown earlier for *wt* GluN2A or GluN2B subunits (*Cummings et al., 1996*; *Prybylowski et al., 2002*; *Thomas et al., 2006*), we turned to autaptic connections, in which hippocampal neurons from *wt* rats are grown at low density (see Materials and methods) that neurons synapse onto themselves (*Figure 5a*), thereby serving as both the presynaptic cell which can be stimulated electrically as well as the postsynaptic cell, from which we can record synaptic currents (*Bekkers and Stevens, 1991*).

Transfected neurons that formed autapses displayed eEPSC$_{NMDA}$ events that were similar in amplitude and kinetics to those seen in non-transfected cells (*Figure 5b* and *Figure 5—figure supplement 1a*), indicating that LiGluN expression makes synaptic NMDA receptors sensitive to light without significantly changing their number, consistent with evidence presented above (*Figure 1—figure supplement 3d*) and with earlier reports which expressed wildtype subunits (*Prybylowski et al., 2002*). In neurons that were transfected with the photo-antagonizing LiGluN2A (G712C) and labeled with L-MAG0, the amplitude of the eEPSC$_{NMDA}$ was inhibited by illumination at 370 nm and this photo-antagonism was relieved by illumination at 510 nm (*Figure 5c, d and f*). The photo-antagonism could be toggled on and off repeatedly (*Figure 5c,d*), whereas non-transfected neurons showed no change in their eEPSC$_{NMDA}$'s amplitude as the wavelength of illumination was switched between 510 nm and 370 nm (*Figure 5e,f*). The magnitude of the light-dependent inhibition of the synaptic eEPSC$_{NMDA}$ was not correlated with eEPSC$_{NMDA}$ amplitude, suggesting that the degree of inhibition varies primarily with labeling efficiency (*Figure 5—figure supplement 1b*; % inhibition ranges from ~20-50%), but also see *Figure 3—figure supplement 1*). Thus, LiGluN subunits incorporate into synaptic NMDA receptors and function normally in response to synaptically released glutamate, while providing for optical control over the synaptic transmission that is mediated by that specific receptor subtype.

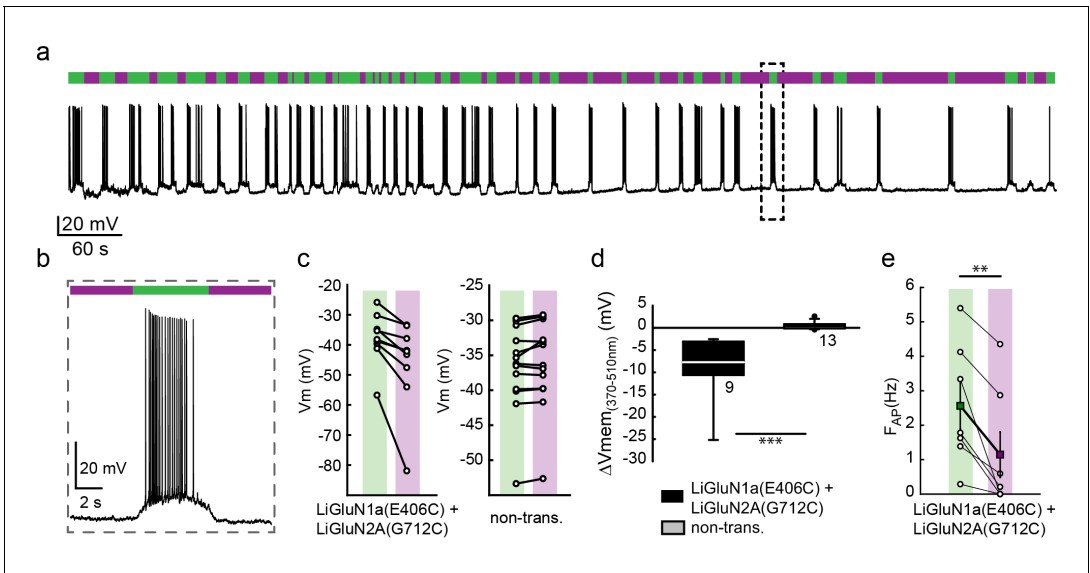

**Figure 4.** Photo-inhibition of neuronal activity with LiGluN1a and LiGluN2A. (**a**) Representative trace showing a long recording (current clamp) of a neuron (from *wt* rats) transfected with LiGluN1a(E406C) and LiGluN2A(G712C), labeled with L-MAG0, and illuminated with 380 nm (violet bars) or 510 nm (green bars) light (held at −40 mV, Mg$^{2+}$-free, 20 μM CNQX). During green light illumination, the neuron displayed strong action potential activity that could be faithfully inhibited by near-UV light (i.e. block is ON), as the photo-antagonism hyperpolarized the cell to suppress action potential firing, (**b**), as typically seen with soluble GluNRs blockers. (**c-d**) Summary of photo-antagonism on membrane potential on individual transfected- and non-transfected neurons (from *wt* rats) (**c**), and summary of the effect on membrane potential (ΔVm) is shown in (**d**). (**e**) Summary for the reduction in firing frequency following violet light illumination of individual cells (circles) and averages ± SEM (color-coded filled squares). Statistics in panel **d** are shown as Box plots of the data (not normally distributed, see *Figure 4—figure supplement 2*), showing median and outliers (filled circles) followed by a nonparametric Mann-Whitney Rank Sum Test (see methods), ***p<0.001, **p<0.01, n.s.- not-significant, *n* shown next to plots. Statistics in panel **e** are shown as mean ± SEM, tested with two-tailed, paired t-test; **p<0.01.

The following figure supplements are available for figure 4:

**Figure supplement 1.** Photo-*antagonism* of NMDA receptors in hippocampal neurons inhibits the development of EPSCs.

**Figure supplement 2.** Summary of nonparametric statistics for *Figure 4*.

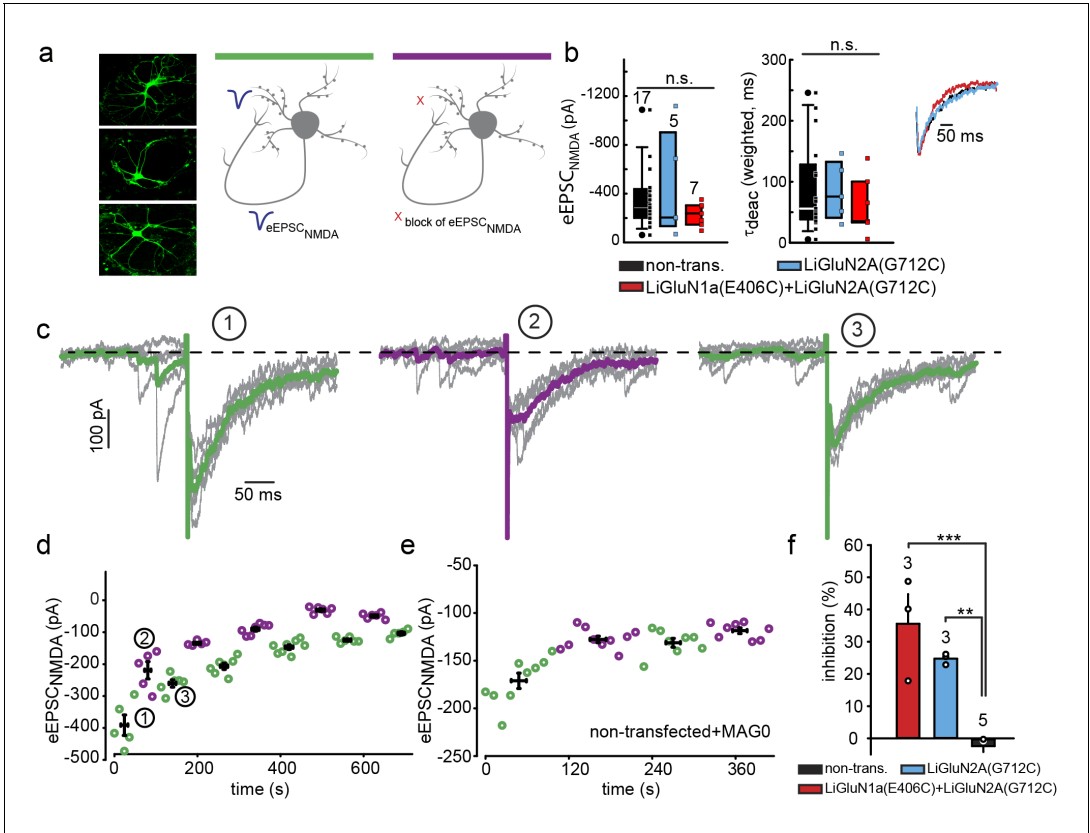

**Figure 5.** Photo-block of synaptic transmission by LiGluNs in hippocampal autapses. (**a**) (left) Representative images of hippocampal neurons (from *wt* rats) grown in low density forming autapses. (right) Schematics of hippocampal autapse used to measure the effect on the NMDA receptors EPSC (blue trace, evoked NMDA EPSC- eEPSC$_{NMDA}$) of turning ON (violet bar) and OFF (green bar) the photo-antagonsim of LiGluN. (**b**) No significant difference in eEPSC$_{NMDA}$ amplitude (left) or deactivation times (bi-exponential weighted $\tau_{deact}$) (right) in autaptic neurons expressing GluN2A(G712C) alone (cyan) or GluN2A(G712C) and GluN1a(E406C) (red) compared to non-transfected control neurons (black) (from *wt* rats). Inset shows superimposed eEPSC$_{NMDA}$ for the three conditions. (**c**) Autaptic NMDA receptor's EPSCs in neurons transfected with GluN2A(G712C) and GluN1a(E406C) before photo-antagonism (1, under 510 nm light). Individual EPSC$_{NMDA}$ (grey) are superimposed with average of 5 consecutive EPSCs (green for 510 nm light; violet for 370 nm light). Photo-antagonism 370 nm light reversibly reduces the amplitude of the eEPSC$_{NMDA}$. (**d**) Time series of NMDA receptors EPSC amplitudes for cell shown in (**c**) reveals repeated reversibility of photo-block of eEPSC$_{NMDA}$, despite the typical rundown of the responses (see [*Goda and Stevens, 1998*]). (**e**) Non-transfected cell exposed to L-MAG0 does not exhibit light-dependent modulation of eEPSC$_{NMDA}$ amplitude. (**f**) Summary of photo-block (as % inhibition) of autaptic NMDA receptors EPSCs in transfected and non-transfected neurons (color scheme as in **b**). 4 cells were excluded (see *Figure 5—figure supplement 1*). Statistics in panels **b** and **c** are shown as Box plots displaying medians, outliers (filled circles) and individual data points (filled, color-coded squares), followed by a nonparametric ANOVA on ranks (see *Figure 5—figure supplement 1*), n.s., not significant. Statistics for panel **f** are shown as mean ± SEM, tested with one way ANOVA, ***p<0.001, **p<0.01, all pairwise Tukey *post hoc* test, *n* shown in parentheses atop bars.

The following figure supplement is available for figure 5:

**Figure supplement 1.** Summary of nonparametric and parametric statistics for *Figure 5*.

## Photo-antagonism blocks LTP-induction in hippocampal slice

An attractive application of optical control over the function of specific NMDA receptors is to probe their function in synaptic plasticity in specific neural circuits. To determine whether such control is possible, we initially turned to the prototypical form of NMDA receptor-dependent plasticity, where tetanic stimulation of hippocampal CA3 Schaffer collateral axons evokes LTP at CA1 pyramidal cells (*Bliss and Collingridge, 1993*; *Lüscher and Malenka, 2012*) (*Figure 6a*). Organotypic slices from

GluN2A-knockout neonate mice (*Sakimura et al., 1995*) were biolistically transfected with GluN1a (E406C), GluN2A(G712C) and tdTomato (typically 1–3 transfected CA1 neurons per slice, *Figure 6b*). Following incubation with L-MAG0, neurons were illuminated with green light (497 nm for 2 s) to place the majority of L-MAG0 in its inactive *trans* state. Then, Schaffer collaterals were stimulated at 0.03 Hz and evoked EPSCs were recorded from CA1 neurons for a 8-minute baseline period. Illumination was then applied for 2 s at either 390 nm (photo-antagonism ON; *Figure 6c*, violet) or 497 nm (photo-antagonism OFF; *Figure 6c*, green). Because of the bistability of MAG (*Figure 1i* and *Figure 1—figure supplement 3e*), LiGluNs remain blocked (following violet light) or unblocked (following green light) in the dark for many minutes. Following the brief illumination protocol, we immediately applied a tetanic stimulation with a pair of 1 s long 100 Hz bursts in the dark. Stimulation was then returned to 0.03 Hz for an extended period to measure changes in synaptic strength (*Figure 6c*). Slices, in which L-MAG0 was maintained in *trans* (photo-antagonism off) by green light, exhibited robust LTP (*Figure 6c*, black circles; EPSC amplitude increased to 185 $\pm$ 31% (n = 10) of baseline, averaged over 20–30 min. post-tetanus; summary in *Figure 6d*), whereas slices in which L-MAG0 was switched to *cis* (photo-antagonism on) by violet light did not (*Figure 6c*, open circles; EPSC amplitude dropped to 67 $\pm$ 28% (n = 10) of baseline, averaged over 20–30 min. post-tetanus; summary in *Figure 6d*). Slices to which L-MAG0 was not added, but which were illuminated at 390 nm, exhibited the same level of LTP as those labeled with L-MAG0 and illuminated with 497 nm light (174 $\pm$ 31.74%, n = 4), demonstrating that the light protocol did not interfere with the generation of LTP and that the block of LTP-induction is a specific outcome of photo-antagonism. Thus, photo-antagonism of post-synaptic NMDA receptors can be used to gate the induction of LTP that is induced by presynaptic bursts of action potentials.

## Single spine GluN2A photo-agonism induces structural plasticity in hippocampal slice

Having shown that precisely timed light pulses can be used to control the activity of synaptic LiGluN containing NMDA receptors, we turned to a second major attraction of light, the ability to focus in small regions of interest, and attempted to control NMDA receptor function at single synapses. Dendritic spine expansion is a structural correlate of LTP, which can be induced at the level of single dendritic spines by the local photo-uncaging of glutamate (*Matsuzaki et al., 2001*; *Matsuzaki et al., 2004*). We asked whether highly local, subunit-specific photo-agonism of GluN2A can induce such structural changes at single spines. As with the experiments, above, which demonstrated the ability to block LTP induction by photo-antagonism of LiGluN2A-containing NMDA receptors, these experiments were performed in organotypic slices obtained from GluN2A-knockout neonate mice (see Materials and methods).

Since calcium entry through NMDA receptors is critical for synaptic plasticity, first we examined the ability of photo-agonized GluN2A(V713C) labeled with L-MAG1 to elicit spine-specific rises in $Ca^{2+}$ in slices co-transfected with the soluble calcium reporter R-GECO1.0 (*Zhao et al., 2011*). Illumination of a single spine head at 405 nm (100 µW for 100 µs/pixel) elicited a rise in R-GECO fluorescence in the illuminated spine, with little or no response in nearby spines or in the dendritic shaft, consistent with spine head $Ca^{2+}$-signaling compartmentalization (*Figure 7b,c*) (*Yuste, 2013*). We next asked whether photo-activation of GluN2A(V713C) can trigger spine expansion. Slices expressing GluN2A(V713C), along with tdTomato as a reporter of spine size (see Materials and methods and [*Holtmaat et al., 2005*]), were labeled with L-MAG1 and imaged in a $Mg^{2+}$-free, high $Ca^{2+}$ solution containing TTX. Spine head illumination with 405 nm (100 µW/µm$^2$ for 600 µs per pixel, ~200 ms per spine) triggered an increase in spine head volume that peaked in ~5 min ($F_t/F_i$ = 1.67 $\pm$ 0.15 of baseline, n = 26, p<0.001, two-tailed, paired t-test) and then declined to an elevated plateau that persisted for at least 45 min ($F_t/F_i$ = 1.31 $\pm$ 0.06 of baseline, n = 26, p<0.001, two-tailed, paired t-test) (*Figure 7d and e*, red symbols). Non-illuminated nearby spines did not change volume (*Figure 7e*, black symbols), nor did illuminated spines in slices that were not labeled with L-MAG1 (*Figure 7d and e*, blue symbols). These experiments show that: i) LiGluN2A(V713C) traffics to synapses, as shown above for LiGluN2A(G712C), ii) photo-agonism of GluN2A(V713C) can be used to induce a spine-specific elevation in internal calcium, and iii) activation of post-synaptic GluN2A-containing receptors is sufficient to induce the morphological correlate of LTP.

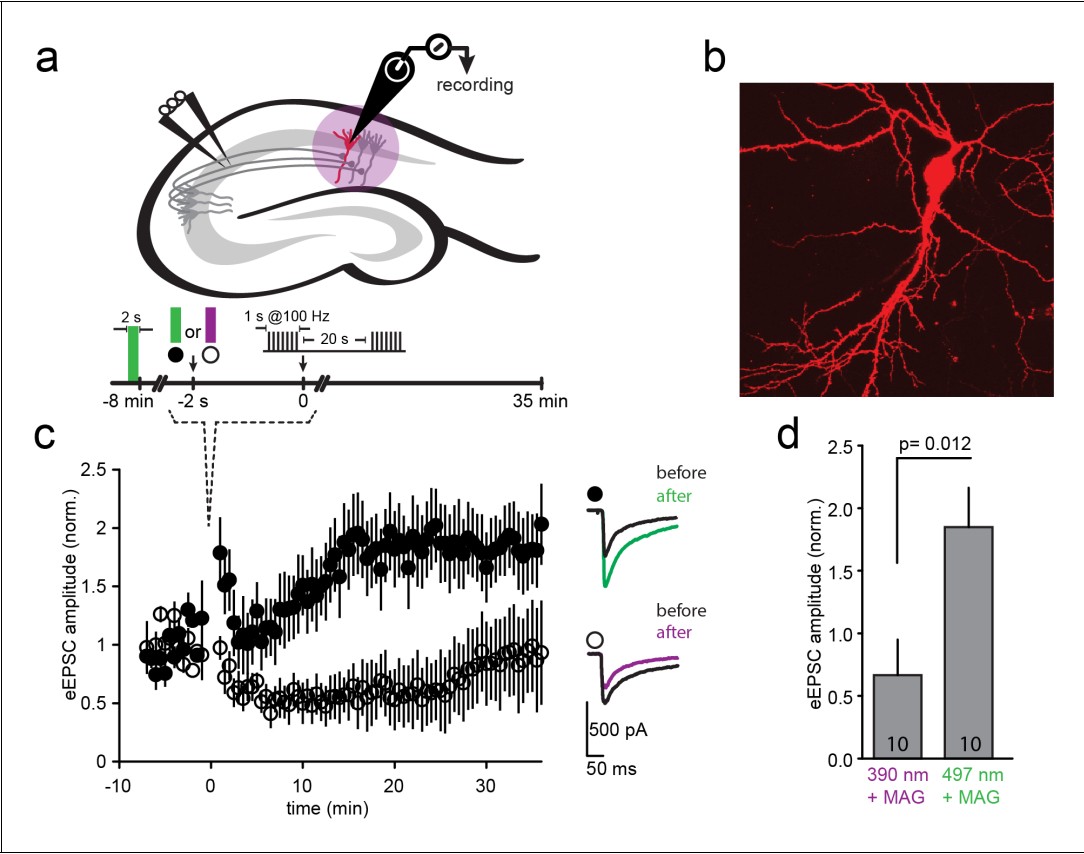

**Figure 6.** LTP induction blocked by photo-*antagonism* of LiGluN1a(E406C) and LiGluN2A(G712C) in organotypic hippocampal slice from GluN2A-knockout neonate mice. (**a**) Schematic of the hippocampus with stimulating electrode on Schaffer collaterals CA3 pyramidal axons that innervate pyramidal neurons of CA1 and recording pipette on a transfected CA1 neuron from Glu2A-KO mice. (**b**) Transfected neuron in an organotypic hippocampal slice identified by tdTomato fluorescence after biolistic co-transfection of LiGluN1a(E406C), LiGluN2A(G712C) and tdTomato. (**c**) Following exposure of the slices to L-MAG0, Schaffer Collaterals were electrically stimulated once every 30 s to obtain a baseline EPSC amplitude (ranging from −8 to 0 min), followed by two high frequency trains (tetanic stimulation: 1-s long trains at 100 Hz, separated by 20 s), followed by a return for 30 min to a one stimulus every 30 s. Normalized mean ± SEM of evoked EPSC amplitudes once every 30 s are shown (ranging from 0 to 35 min). When the tetanic stimulation was preceded by illumination at 497 nm (photo-antagonism OFF; green bar, filled symbols, n = 10) EPSC amplitude approximately doubled, as common for CA3-CA1 LTP. However, illumination at 390 nm (photo-antagonism ON; violet bar, open symbols, n = 10) prevented the generation of LTP. Inset shows representative average EPSCs before (black traces) and after (at t= 20–30 min) the tetanic stimulation (green and violet traces). (**d**) Summary of the results shown in (**c**), averaged between t = 20–30 min. Statistics in panels are shown as mean ± SEM, tested with two-tailed, unpaired t-test, p as indicated, n shown within bars.

## Single spine photo-antagonism gates structural plasticity in hippocampal slice

We next asked whether photo-antagonism of GluN2A and GluN1a could be used to prevent gluta-mate-induced spine expansion at single dendritic spines in organotypic slices obtained from GluN2A-knockout neonate mice. Caged-glutamate (MNI-glu) was added to a $Mg^{2+}$-free bathing solution for 1–2 min followed by photo-uncaging with 405 nm light at a spot of the same lateral size as the spine head, but located 0.5 µm away (*Figure 8—figure supplement 1a*). The 405 nm illumination was well-confined (*Figure 8—figure supplement 1a*) and reliably induced spine expansion (*Figure 8—figure supplement 1b*) when LiGluN2A was transfected, as shown earlier for GluN2A-*wt* (*Lee et al., 2009*; *Matsuzaki et al., 2001*; *Matsuzaki et al., 2004*), but stable expansion did not occur in slices that were not transfected with LiGluN2A (*Figure 8—figure supplement 1c*). To deter-mine if the spine expansion that is triggered by glutamate uncaging could be prevented by photo-antagonism of LiGluN (*Figure 8a*), we co-expressed LiGluN2A(G712C) and LiGluN1a(406C) with tdTomato and labeled the slices with L-MAG0. Individual spines received two successive bouts of

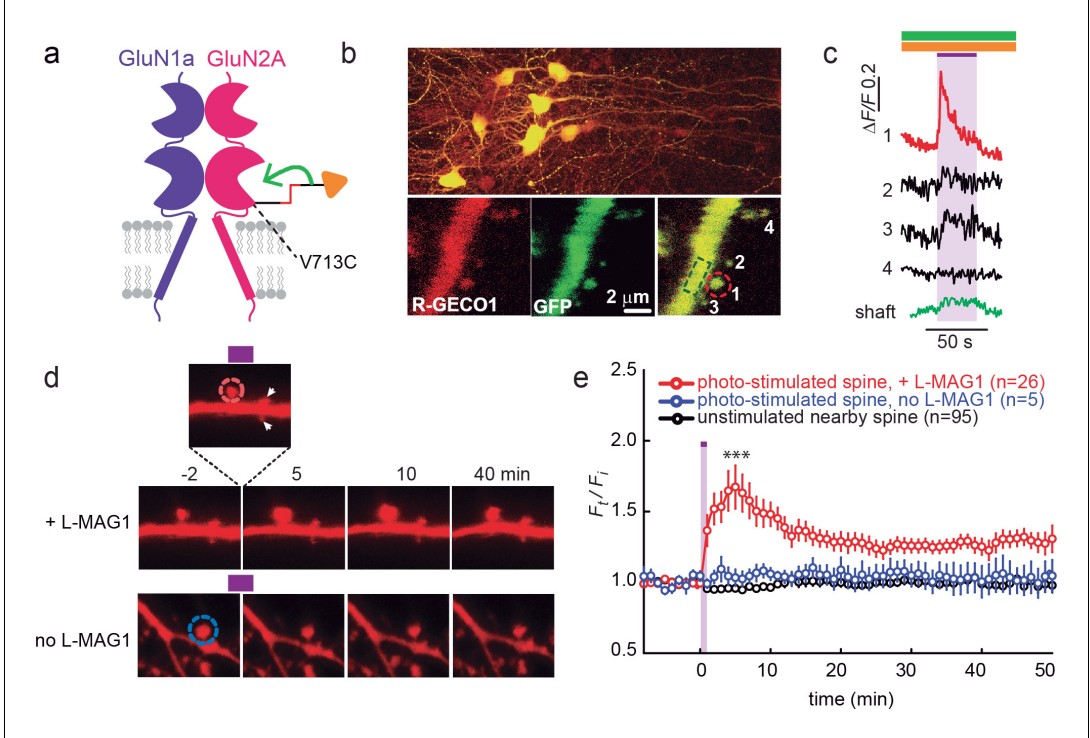

**Figure 7.** GluN2A(V713C) photo-*agonism* triggers calcium increase and expansion in single dendritic spines of organotypic hippocampal slice from GluN2A-knockout neonate mice. (a) Schematic of photo-agonism with L-MAG1 conjugated to the LiGluN2A(V713C) subunit. (b, c) Single-spine calcium rise induced by photo-agonism. (b) CA1 neurons co-expressing LiGluN2A(V713C) fused to R-GECO1.0 [LiGluN2A(V713C)-R-GECO1.0] and soluble R-GECO1.0 (red) as well as GFP (green) in merged low power image showing cluster of transfected neurons (top) and high-magnification image of dendritic shaft with several spines, showing the green, red and merged channels (bottom). Photo-agonism with 405 nm illumination focuses on spine #1 (red dashed circle), while R-GECO1.0 imaging measures calcium in spines # 1–4 and the dendritic shaft (dashed green rectangle). (c) Illumination of spine #1 in (b) at 405 nm (violet bar, with additional excitations at 488 and 561 nm to image eGFP (green bar) and R-GECO1.0 (orange bar), respectively, elicits an increase in R-GECO1.0 fluorescence within the head of spine #1 (red trace), while other spines (black traces) and the dendritic shaft (green trace) display little or no increase in fluorescence. (d) Single spine expansion by photo-agonism of LiGluN2A. Spines from CA1 pyramidal neuron co-expressing GluN2A(V713C) and tdTomato and treated with L-MAG1 were stimulated with repetitive 405 nm laser scanning at the tip of the spine head, at ~1 Hz for a total amount of 1–2 min. (Top) Time lapse of a representative spine undergoing expansion after on-spine stimulation with 405 nm light (violet bar on top of inset, dashed red circle). Non-illuminated, nearby spines (white arrowheads) do not exhibit any change in volume. (Bottom) Representative spine from a slice without L-MAG1 treatment showing no response to 405 nm light (blue dashed circle). Numbers above images indicate time in min relative to photo-stimulation at $t = 0$. (e) Summary of results. Statistics in panel (e) are shown as mean ± SEM, \***p<0.001, two-tailed, paired t-test compared to baseline, $\Delta t$= -10–0 mi. $n$, from series of experiments as in (d), is shown in parentheses.

near-spine glutamate photo-uncaging, the first following induction of LiGluN photo-antagonism by 405 nm on the spine head (100 µW/µm$^2$ for 600 µs per pixel), and the second following reversal of the photo-antagonism by 488 nm light (100 µW/µm$^2$ for 150 µs per pixel) (*Figure 8b,c*). In this experimental design, the second bout of uncaging was crucial to determining that a spine that did not respond to the first bout of uncaging was nevertheless competent to expand, asmany spines are not (*Lynch et al., 2013*). The first uncaging bout, during GluN1a/GluN2A photo-antagonism, induced only a small, transient increase in spine head volume, which peaked in 1 min ($F_t/F_i$ = 1.18 ± 0.07 of baseline, n = 12, p>0.1, two tailed, paired t-test) and shrank back to near baseline ($F_t/F_i$ F = 1.05 ± 0.05 of baseline, n = 12, p>0.1, two tailed, paired t-test) within 5 min (*Figure 8c*, red symbols), indicating a lack of the long-lasting expansion that is associated with LTP (*Yang et al., 2008*). After turning off the GluN1a/GluN2A photo-antagonism with green light, the second bout of near-spine glutamate uncaging induced a large persistent expansion ($F_t/F_i$ = 1.47 ± 0.13 compared to new moderately elevated baseline between t = 3 min and 15 min, n = 12, p<0.01, two tailed, paired t-test) (*Figure 8c*, red symbols). Nearby, non-illuminated spines did not respond (*Figure 8b and c*, black symbols). These experiments show that photo-antagonism of GluN1a/GluN2A can block the

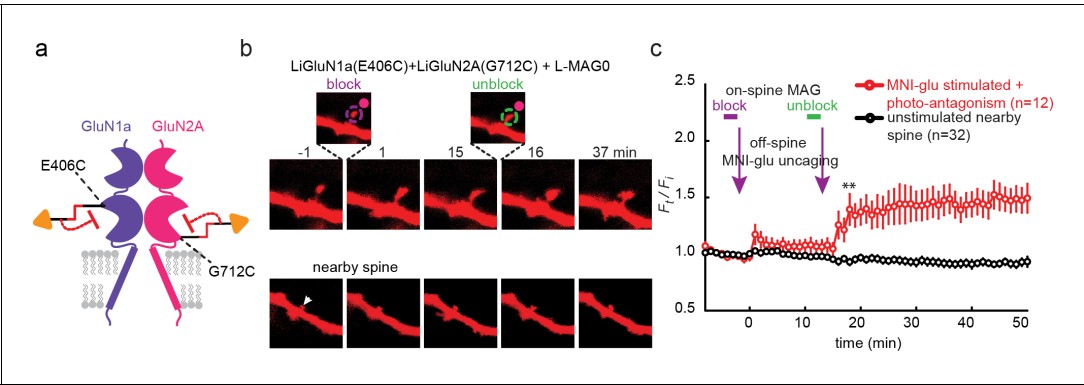

**Figure 8.** Photo-*antagonism* prevents glutamate-induced expansion in single dendritic spines of organotypic hippocampal slice from GluN2A-knockout neonate mice. (**a**) Schematic of a photo-*antagonized* NMDA receptors containing LiGluN1a(E406C) and LiGluN2A(G712C) conjugated to L-MAG0. (**b**) CA1 pyramidal neuron co-expressing LiGluN1a(E406C), LiGluN2A(G712C) and tdTomato following treatment with L-MAG0. (Top) Single spine receiving two bouts of near-spine glutamate-uncaging at 405 nm (magenta spots, insets), the first after photo-antagonism was induced by 405 nm on-spine illumination (violet dashed circle, left inset), the second following removal of the photo-antagonism by 488 nm on-spine illumination (green dashed circle, right inset). Photo-block (and relief of block) of the receptors was done by stimulating the tip of the spine head with repetitive 405 nm (or 488) laser scanning, at ~1 Hz for a total amount of 1 min. MNI-glutamate uncaging was performed slightly away from the spine head (~2 μm) to avoid further stimulation of the receptors found on the spine head. The spine undergoes a small expansion in response to the first glutamate-uncaging (p=0.06, compared to baseline at t= -8 – 0 min), and a large expansion following removal of the photo-antagonism. (Bottom) A nearby non-illuminated spine (arrowhead) does not change size. Numbers above images indicate time in min relative to photo-stimulation at t = 0. (**c**) Summary of results from a series of experiments as in (**b**) (mean ± SEM, **p< 0.01, two-tailed, paired t-test compared to baseline at t= 9–15 min, n shown in parentheses).

The following figure supplement is available for figure 8:

**Figure supplement 1.** Single spine spatial precision of photocontrol in organotypic hippocampal slice from GluN2A-knockout neonate mice.

induction of spine expansion and that this block is reversible, enabling a morphological correlate of LTP to be gated temporally at single synapses.

## LiGluN2A photo-antagonism disrupts pruning of zebrafish retinal ganglion cell axonal arbor in vivo

Having seen that LiGluNs provide optical control over NMDA receptor dependent transmission and plasticity in dissociated neurons and in brain slice, we turned to test their utility in vivo in larval zebrafish, *Danio rerio*. As a model of NMDA receptor-dependent plasticity, we examined the synaptic pruning that takes place in the developing visual system. NMDA receptors play an important role in the formation of sensory topographic maps (*Bliss and Collingridge, 1993*; *Lüscher and Malenka, 2012*). In many vertebrates, including zebrafish, exposure to soluble NMDA receptor antagonists during a critical period in development of retinotectal projections can disrupt the convergence of retinal ganglion cell axons, thus disrupting retinotopy (*Cline and Constantine-Paton, 1989*; *Ruthazer et al., 2003*; *Schmidt, 1991*; *Zhang et al., 1998*). In zebrafish larvae, one such critical period occurs between 5 and 7 days post fertilization (dpf), coincident with the onset of behaviors that require visual acuity, such as prey capture. Larvae exposed to soluble NMDA receptor antagonists during this critical period exhibit enlarged RGC axonal arbors (*Schmidt et al., 2000*).

We asked whether photo-antagonism of LiGluNR2A(G712C) during the 5–7 dpf critical period would affect retinal ganglion cell axon growth. We generated a transgenic line of zebrafish expressing GluN2A(G712C) under the control the gal4-UAS expression system (*Scott et al., 2007*). Double transgenic Tg[*s1101t-gal4;UAS-GluN2A(G712C)*] fish were incrossed to produce embryos with GluN2A(G712C) expressed throughout the nervous system (*Figure 9b*), including the retina and optic tectum. To visualize individual retinal ganglion cells, embryos were injected with DNA for GFP expression (see methods) to sparsely mark a subset of retinal ganglion cells projecting to the optic tectum (*Figure 9b*, right panels) (*Xiao and Baier, 2007*). At 5 dpf, healthy larvae expressing LiGluN2A(G712C) pan-neuronally, with sparse expression of GFP in retinal ganglion cells, were divided into three treatment groups: (1) L-MAG0, (2) vehicle (DMSO only), or (3) MK-801. All larvae

were mounted in agarose to image baseline axon arbor morphology, and then freed and transferred to a 48-well plate imaging chamber (*Levitz et al., 2013*). Between 5 and 7 dpf, freely swimming larvae were exposed to a 10 s flash of 405 nm light once every 30 min (*Figure 9a*). Because MAG is bistable (see *Figure 1*), we reasoned that brief bouts of isomerization into the photo-antagonizing state would be sufficient to provide long-term antagonism, while allowing the larvae to develop under predominantly natural visual stimuli. At 7 dpf, larvae were imaged a second time to assess the growth of each axon arbor.

The size of a retinal ganglion cell axon arbor was determined by a 3-dimensional Sholl analysis (*Figure 9c*), which compared z-stacks obtained at 5 dpf and 7 dpf. At 5.25 dpf (126 ± 2 hr post fertilization), the size distribution of retinal ganglion cell axon arbors from Tg[*s1101t-gal4;UAS-GluN2A (G712C)*] larvae was indistinguishable that of siblings lacking the *UAS-GluN2A(G712C)* transgene (*Figure 9d*) (p>0.7, two- two-tailed, unpaired t-test, n = 19 Tg[*s1101t-gal4;UAS-GluN2A(G712C)*] and n = 10 control axons), indicating that gal4-driven expression of UAS-GluN2A(G712C) does not alter the initial development of the retinotectal projection. However, by 7 dpf, axon arbors in Tg[*s1101t-gal4;UAS-GluN2A(G712C)*] larvae that were illuminated episodically to induce photo-antagonism showed a significant increase in arbor radius (*Figure 9f*, ΔR = 16.4 ± 1.7%, n = 12 axons), compared to illuminated vehicle-treated (ΔR = 1.2 ± 3.4%, n = 9 axons) or wild-type siblings (ΔR = 6.9% ± 2.6%, n = 10 axons) (*Figure 9—figure supplement 1a*). Tg[*s1101t-gal4;UAS-GluN2A(G712C)*] larvae that were not treated with MAG, but were instead reared in 25 μM MK-801, under the same conditions from 5–7 dpf, showed a comparable increase in arbor radius (ΔR = 13.8 ± 3.7%, n = 10 axons), indicating that the efficacy of photo-antagonism was on par with a relatively strong dose of a soluble NMDA receptor blocker. MAG treatment alone had no effect on development compared to vehicle-treated siblings, based on an assessment of spontaneous swim behavior, escape behavior and habituation of escape behavior (*Figure 9—figure supplement 1b* and Materials and methods). These results demonstrate that extended photo-antagonism of LiGluN2A(G712C) mimics the effect of in vivo chronic systemic exposure to a soluble NMDAR antagonist, producing a block of activity-dependent remodeling of retinotectal projections.

## Discussion

We have used chemical optogenetics to engineer light-controlled NMDA receptors that can be used to study the mechanisms of NMDA receptor function and of synaptic transmission and plasticity. We generated a family of four light-gated NMDA receptor-subunits that provide reversible spatiotemporal control of receptor activity and synaptic plasticity *via* a photochemical switch that either activates or antagonizes specific 'LiGluN' subunits: LiGluN1a, LiGluN2A and LiGluN2B. The photo-agonized subunits can be photoactivated to yield persistent currents or be sculpted in time to mimic synaptic activation of the receptors (*Figure 1*). Photo-antagonism blocks the effects of perfused ligands (glutamate/NMDA) and of synaptically released glutamate, thus controlling NMDA receptor mediated EPSCs and LTP. Spatial confinement of photo-antagonism to a single dendritic spine blocks the spine expansion that is associated with LTP (*Figures 2–6* and *8*). As a complement to this, single-spine photo-agonism evokes spine-specific $Ca^{2+}$-transients and spine expansion (*Figure 7*).

We also report a transgenic zebrafish line that expresses the light-antagonized version of GluN2A, which enables chronic antagonism over days of larval development after a single application of MAG (*Figure 9*). We take advantage of the bistability of MAG (*Figure 1i* and *Figure 1—figure supplement 1e*) to produce chronic photo-antagonism with short pulses of light at long intervals over a period of 40 hr in freely swimming fish, thereby minimizing visual stimulation and avoiding photo-toxicity. We find that the photo-antagonism is as potent as MK-801 in interfering with the normal NMDA receptor dependent pruning process, producing an overgrowth of retinal ganglion cell axons in the optic tectum. Conveniently, in the zebrafish system, MAG can simply be added to the water to be taken up systemically, as shown earlier (*Janovjak et al., 2010*; *Levitz et al., 2013*; *Szobota et al., 2007*; *Wyart et al., 2009*).

For photo-controlled NMDA receptors to work under physiological expression conditions, one would want LiGluN subunits to replace native subunits, rather than increasing the pool of NMDA receptors. Fortunately, in some preparations, GluN2A overexpression does not appear to increase synaptic content of NMDA receptors (*Barria and Malinow, 2002*; *Prybylowski et al., 2002*), most likely because of the limiting supply of the native obligatory partner GluN1 subunit. We find that

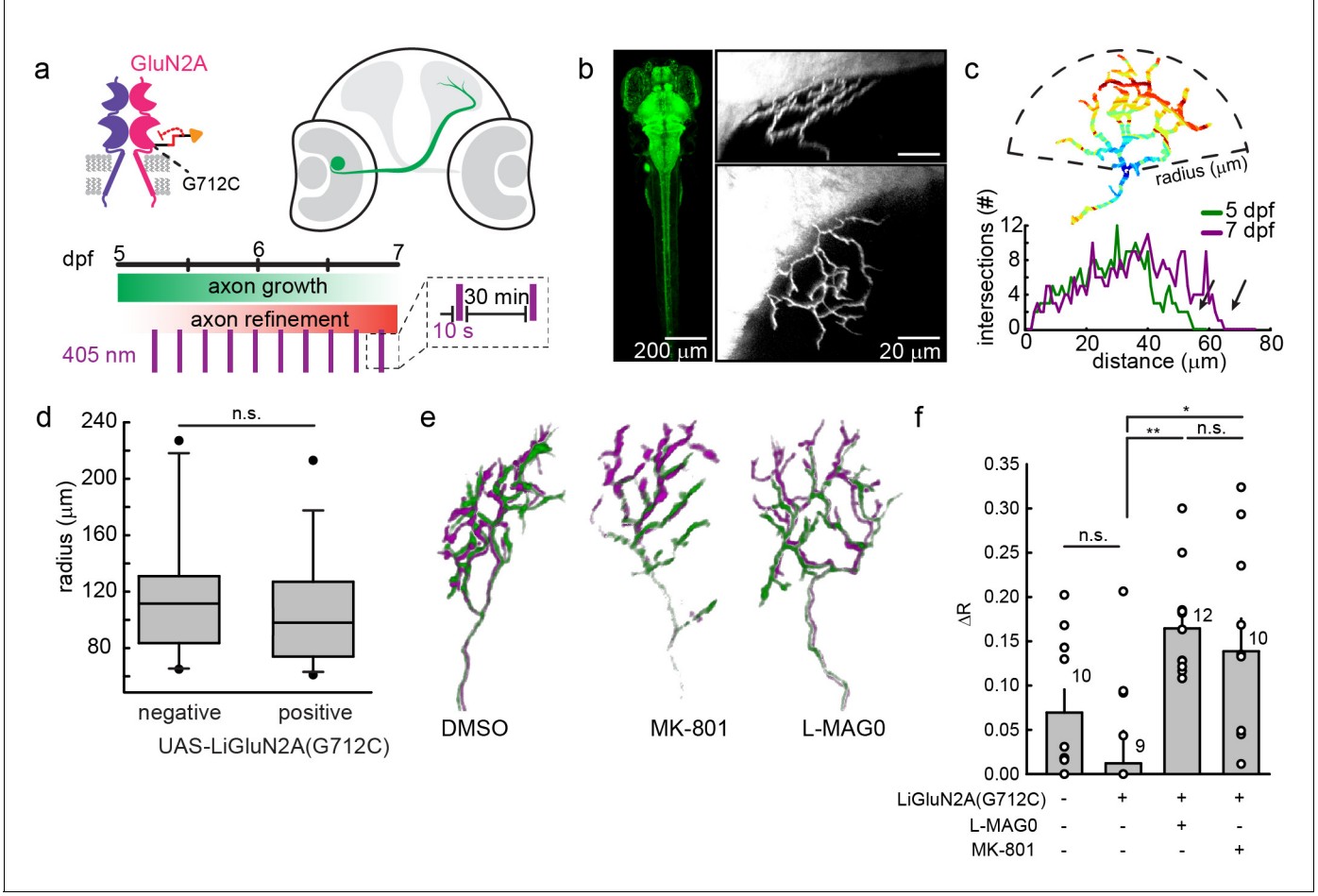

**Figure 9.** GluN2A(G712C) photo-antagonism disrupts refinement of retinal ganglion cell axon arbors in larval zebrafish in vivo. (**a**) Cartoons depicting GluN2A(G712C) photo-antagonism (top left), development of retinal ganglion cell projection (top, right) and timeline of the photo-antagonism assay (bottom). (**b**) (Left) Dorsal view of 5 dpf larva showing pan-neuronal expression pattern driven by *s1101t-gal4* visualized by expression of *UAS-GCaMP3*. Transverse (right, top) and dorsal (right, bottom) tectal projection of a retinal ganglion cell axon arbor labeled by mosaic expression of *pou4f3:mGFP* at 7 dpf. (**c**) Axon arbors were traced and arbor radius measured by a 3-dimensional Sholl analysis counting the number of intersections encountered by concentric spheres centered on the first branch point. Arrows indicate arbor radius (R) at 5 dpf and 7 dpf. (**d**) Prior to antagonism, Tg[*s1101t-gal4; UAS: GluN2A(G712C)]* animals have a comparable distribution of arbor radii at 5 dpf as non-expressing animals (without UAS, but mix of s1101t +/-) (n.s., p >0.6, Mann-Whitney Rank Sum Test, n = 19, GluN2A(G712C)-expressing axons and 10 non-expressing axons). Box plot whiskers indicate 5% and 95% percentiles. Dots above whiskers represent outliers. (**e**) Representative selection of retinal ganglion cell axon arbors in transgenic animals at 5 dpf (green) and 7 dpf (magenta). Larvae were treated at 5 dpf with either 150 µM L-MAG0 in 0.3% DMSO, 0.3% DMSO alone, or 25 µM MK-801, and subjected to 10 s flashes of 405 nm light at 30 min intervals from 128–168 hpf. (**f**) In animals where GluN2A(G712C) was photo-antagonized by MAG, retinal ganglion cell axon arbors grew significantly more compared to DMSO-treated control groups (left-to-right: ΔR = 6.9%, n = 10 axons, ΔR = 1.2%, n = 9 axons; ΔR = 16.4%, n = 12 axons). This overgrowth was comparable to the change in arbor radius observed in animals treated with MK-801 (ΔR = 13.8%, n = 10 axons). For statistical analysis of the relative change in arbor radius (ΔR = (R$_{7dpf}$-R$_{5dpf}$)/R$_{5dfp}$), we initially compared untreated –*wt* animals and untreated transgenic animals (two left bars) and observed no statistical difference (mean ± SEM, n.s. not significant, two-tailed, unpaired t-test). This enabled the comparison between the animals from the same transgenic background (second to fourth bar) for the effect of photo-inhibition on radius (mean ± SEM, n.s. not significant, **p<0.01, *p<0.05, tested with one way ANOVA, all pairwise Tukey *post hoc* test (see *Figure 9—figure supplement 1c*). Individual data points are shown for **f** (open circles).

The following figure supplement is available for figure 9:

**Figure supplement 1.** MAG treatment of zebrafish larvae in the absence of LiGluN2A expression does not affect behavior.

expression of LiGluN2A makes NMDA receptors sensitive to light without altering whole cell NMDA-induced current or NMDA receptor EPSCs evoked by action potential-driven glutamate release at autaptic synapses. The degree of photo-block of NMDA receptor EPSCs in autapses from neurons from *wt* rats (*Figure 5*) is small on average (mean 25–35%, range ~20–50% without inclusion of non-responding cells, see *Figure 5—figure supplement 1*), but shows that functional light-controlled receptors can be obtained in a wild-type background where the LiGluN subunits compete with *wt* subunits to assemble into receptors that are properly trafficked to the plasma membrane of the synapses. It should be noted that, the expression of the photoswitchable subunits in a wild-type background where the native subunit is expressed yielded variable effects, ranging from very strong block to no block at all (*Figure 5—figure supplement 1b*). This may result from a combination of factors, including variation in expression level that results in variable incorporation into GluN2-containing receptors. Nevertheless, in responding cells, regardless of the magnitude of fractional optical control, the control is reproducible (*Figure 3—figure supplement 1* and *Figure 5—figure supplement 1b*). The degree of photo-block in other preparations, such as organotypic slices from the GluN2A-KO mice, where we demonstrated block of LTP (*Figure 6*) and either induction or block of spine expansion (*Figures 7–8*), is likely greater because of absence of the native GluN2A subunit.

The chemical optogenetic approach is selective for the subunit that bears the cysteine anchoring site for the MAG photoswitch. Thus, our toolset allows for the direct and specific optical manipulation of GluN2A or GluN2B receptors and should make it possible to address the roles of these subunits in synaptic plasticity, circuit function and memory (*Foster et al., 2010*; *Kohl et al., 2011*; *Liu et al., 2004*; *Massey et al., 2004*; *Sakimura et al., 1995*; *Weitlauf et al., 2005*). In addition, owing to the ability to localize and shape light, it should also be possible to determine how synaptic transmission and circuit function are shaped by receptor type at various subcellular locations and within single dendritic spines.

Chemical labeling and azobenzene isomerization have limitations that need to be considered. While neurons in culture and slice are easily incubated with MAG, and MAG can be readily delivered systemically in zebrafish when added to the swimming media (*Figure 9*) (*Janovjak et al., 2010*; *Levitz et al., 2013*; *Szobota et al., 2007*; *Wyart et al., 2009*), we do not know the efficiency of this process and incomplete conjugation (i.e. partial occupancy) may contribute to the incomplete photo-agonism and photo-antagonism that we observe. In addition, a small fraction of MAG remains in *trans* during 380 nm illumination (*Gorostiza et al., 2007*), and there may be differences in occupancy in the *cis* state of MAG molecules that are bound to the cysteine at the two possible stereochemistries of attachment to the maleimide (*Figure 1a*) (*Numano et al., 2009*). We have recently solved two of these challenges for photoswitching a metabotropic glutamate receptor by replacing the maleimide-cysteine conjugation with the bio-orthogonal and very efficient conjugation of a benzylguanine photoswitch to a SNAP tag fused to the N-terminal of the LBD (*Broichhagen et al., 2015*).

Because the cysteine-reactive maleimide of MAG hydrolyzes, receptor conjugation will occur in the first hours after addition of MAG. Only those receptors that are on the plasma membrane during this time window will be labeled and become photo-controllable. As a result, the time frame of an experiment is limited by receptor-turnover, and for in vivo experiments extending several days, MAG may need to be reapplied. It also means that for studies of development, cells born after MAG application will not be under light control. This will mean that the tools do not work for some applications, or require an extra step (i.e. MAG reapplication). In some cases it could actually provide an advantage to selectively antagonize or agonize based on receptor 'birthdate'.

Recent advances in the design of azobenzene-based photo-switches have yielded new classes of MAG (*Kienzler et al., 2013*) that exhibit favorable two-photon absorption (*Carroll et al., 2015*; *Gascón-Moya et al., 2015*; *Izquierdo-Serra et al., 2014*). This is particularly advantageous for light confinement (*Emiliani et al., 2015*), especially for small subcellular compartments, and for deeper tissue penetration, for in vivo applications. Fortunately, in preparations such as *C.elegans*, *Drosophila* and zebrafish, where precise genetic targeting can be readily achieved (i.e. of the photoswitch-ready subunit) and where the nervous system is accessible to light and less scattering, it is already possible with conventional 1-photon illumination to accomplish MAG photo-control of glutamate receptors, as previously shown in zebrafish sensory neurons (*Szobota et al., 2007*), the central pattern generator circuit for swimming (*Janovjak et al., 2010*; *Wyart et al., 2009*), pan-neuronally (*Levitz et al.,*

*2013*), and as shown here in transgenic zebrafish expressing the LiGluN2A(G712C) subunit (*Figure 9*).

In conclusion, we have engineered a palette of light-controlled NMDA receptor subunits, which enable the fast and reversible remote control of specific receptor subtypes, in specific cells and thereby with presynaptic versus postsynaptic selectivity. These properties should enable a new level of study of the molecular mechanism of function of NMDA receptors and how specific NMDA receptors participate in synaptic transmission, integration and plasticity.

## Materials and methods

### Site-directed mutagenesis
Cysteine point mutations were introduced near, but outside of the glutamate-binding site of GluN2A, GluN2B and GluN1a using site-directed mutagenesis and verified by full sequencing.

### Cell culture and transfection
HEK293 or HEK293T cells were maintained in DMEM with 5% FBS on poly-L-lysine-coated glass coverslips at ~3 x $10^6$ cells per milliliter and transiently co-transfected with GluN1a and GluN2A plasmids at a DNA ratio of 1:2 using Lipofectamine 2000 (Invitrogen/Life technologies, Carlsbad, CA). Calcium imaging or patch clamping was performed 12–48 hr after transfection. Dissociated postnatal hippocampal neurons (P0-P5) were prepared from Sprague Dawley rats (Charles River) at high density (80 K cells/coverslip) and transfected as described previously (*Berlin et al., 2015*; *Szobota et al., 2007*), using 1 μg of cDNA encoding the NMDA receptor subunit(s) and 0.2 μg of GFP. For autaptic recordings, cells were plated at a lower density (20 K cell/coverslip) with the extracellular medium enriched with 15 mM KCl, to promote the formation of autaptic connections. As an additional precaution, we transfected these neurons at very low efficiency so that very few neurons in a dish expressed LiGluN. We took this additional provision so that the illumination would only affect the population of LiGluN receptors located on the recorded cell, without affecting the activity of other cells in the vicinity (of which the majority are non-transfected even under normal transfection methods nonetheless), as is the case of other soluble blockers. For *Xenopus* oocyte mRNA injection, we have cloned GluNRs and LiGluNRs into pGEM-HJ, synthesized mRNA in vitro and injected into defolliculated oocytes; as previously described (*Berlin et al., 2010*).

### Organotypic hippocampal slice culture preparation and biolistic transfection
All experiments on hippocampal slices (physiological experiments on LTP induction and single spine expansion experiments) were in cultured (organotypic) slices from GluN2A-knockout mice (*Sakimura et al., 1995*). Briefly, slices (400 μm thick) were prepared at postnatal days 6–8 as previously described (*Fuller and Dailey, 2007*). Slices were cut in ice-cold, oxygenated dissection solution (95% $O_2$/5% $CO_2$) containing (in mM) 1 $CaCl_2$, 10 dextrose, 4 KCl, 5 $MgCl_2$, 26 $NaHCO_3$, 233 sucrose, 0.0005% phenol red, and grown on cell culture inserts (Millipore, EMD) in culture medium consisting of neurobasal medium (no L-glutamine, GIBCO), horse serum (Hyclone), insulin (Sigma), ascorbic acid (Sigma), Glutamax (GIBCO), penicillin-streptomycin (GIBCO) and HEPES (pH 7.4) at 34°C. Medium was supplemented with 4 μM cytosine β-D-arabinofuranoside hydrochloride (AraC, Sigma) the day after. Slices were biolistically transfected with gold particles (1 μm, Bio-Rad) covered with the appropriate DNA combination (*O'Brien and Lummis, 2006*). AraC was withdrawn from the media prior to transfection and 10 μM MK-801 (Tocris) was added at that time. The MK-801 was removed on the day of experiment.

### Calcium-imaging in HEK293 cells
HEK293 cells were loaded in recording solution with 5 μM Fura2-AM (Molecular Probes) for 30 min at 37°C, 5% CO2. The recording solution contained (in mM): 150 NaCl, 5 KCl, 0.2 $CaCl_2$, 10 D-glucose, 10 D-sucrose, 10 HEPES, 0.01 EDTA, 0.05 glycine, pH 7.4. Cells were labeled with 50–100 μM MAG in glycine-free recording solution. Changes in intracellular [$Ca^{2+}$] in individual cells were measured from Fura2-AM fluorescence intensity by brief (<1 s, ~1.5 μW/mm$^2$) excitation at 350 nm and 380 nm at 5 s intervals and by detecting emission at 510 nm.

## Electrophysiology

Patch clamp recordings used an Axopatch 200 A amplifier in the whole cell mode. Recordings were carried out 12 to 48 hr after transfection in HEK cells and after 15 DIV for hippocampal neurons. Cells were pre-treated with 1 mM DTT for 5 min, rinsed for 10 min, and incubated with 50–100 µM MAG (and for GluN2A(G712C), 500 µM AP5) for 30 min at 37°C, 5% $CO_2$. The labeling solution contained (in mM): 150 NMDG-HCl, 3 KCl, 0.5 $CaCl_2$, 5 $MgCl_2$, 10 HEPES, 5 D- glucose, pH 7.4. Cells were voltage-clamped at -60 mV. Pipettes had resistances of 2–8 MΩ and were filled with a solution containing, for HEK cells (in mM): 110 D-gluconic acid, 30 CsCl, 4 NaCl, 5 HEPES, 5 BAPTA, 0.5 $CaCl_2$, 2 $MgCl_2$, pH 7.3, and for neurons (in mM): 136.5 K-gluconate, 17.5 KCl, 9 NaCl, 1 $MgCl_2$, 10 HEPES, 0.2 EGTA, pH 7.3. The extracellular recording solution for HEK cells was as described above for calcium-imaging experiments. For dissociated rat hippocampal neurons, when assessing the relative photo-current (or block) compared to the total NMDA- induced current the extracellular recording solution, nominally $Mg^{2+}$-free and with high external $Ca^{2+}$, contained (in mM): 138 NaCl, 1.5 KCl, 10 D-glucose, 3.7 $CaCl_2$, 5 HEPES, pH 7.4 and 0.05 glycine. All other experiments with cultured neurons were performed with 2.5 mM external $Ca^{2+}$. Illumination was applied using a Polychrome monochromator (TILL Photonics) (also see below) coupled to the back port of an Olympus IX70 inverted microscope. Light intensity measured at the 40x objective was ~3 mW/mm$^2$.

Recording of autapses was performed using a standard protocol as described in (*Levitz et al., 2013*). Briefly, cells we held at −70 and depolarized to +20 or 40 mV for 3–5 ms, then returned to −70 mV. For recording NMDA-dependent spontaneous or evoked EPSC (sEPSC$_{NMDA}$ and eEPSC$_{NMDA}$, respectively), cells were incubated with 20 µM CNQX, in nominally $Mg^{2+}$-free extracellular recording solution and with 2.5 mM $CaCl_2$. To note, during autaptic recordings we consistently observed the gradual reduction in eEPSC$_{NMDA}$ amplitude (transfected and non-transfected neurons), consistent with previous reports (*Goda and Stevens, 1998*).

Hippocampal slices were obtained from GluN2A-knockout neonate mice and electrophysiological recordings to measure LTP induction were done on at 6–8 d in vitro. Just before recording, slices were incubated at room temperature (~25°C) with 100 µM TCEP for 1 min, rinsed for 2 min, and incubated for 45 min with 250 µM MAG and 500 µM AP5 diluted in the NMDG-labeling solution (see above). Slices were rinsed twice in labeling solution before recording. Whole-cell patch-clamp recordings were performed on an upright Zeiss AxioExaminer using an Axopatch 200B amplifier (Molecular Devices). A bipolar stimulating electrode was placed along the Schaffer collateral pathway and post-synaptic currents were recorded in whole-cell mode from transfected CA1 pyramidal neurons (*Figure 6*). The internal solution contained (in mM): 142 CsCl, 2 $MgCl_2$, 1 EGTA, 10 HEPES, 0.4 Na3GTP, 4.4Na2ATP, 5 QX314, pH 7.4. Slices were perfused with a medium containing (in mM): 118.9 NaCl, 2.5 KCl, 2 NaH2PO4, 26.2 NaHCO3, 11 D-Glucose, 2.5 CaCl2, 1.3 MgCl2 and 0.01 glycine, pH 7.4 when saturated with 95% $O_2$ / 5% $CO_2$. The light used for photoswitching was from a DG-4 (Sutter Instruments) coupled to the microscope and projected onto the sample through a 40× objective. Light intensity measured at the sample was approximately 43 mW/mm$^2$ at 390 nm and 51 mW/mm$^2$ at 497 nm. EPSCs were recorded in voltage-clamp mode and the tetanus (consisting of two 1 s trains of 100 Hz separated by 20 s) was delivered in current-clamp mode.

Two-electrode voltage clamp experiments were performed in *Xenopus laevis* oocytes injected with 50 nl mRNA of the different GluN subunits at 1–1.5 ng/oocyte. GluN2A (*wt*, G712C, V713C) and GluN2B (*wt*, V714C) were injected with GluN1a-*wt* at a ratio of 1:1. GluN1a (E406C) was injected with GluN2B-*wt*. Cells were then incubated in ND-96 (96 mM NaCl, 2 mM KCl, 1.8 mM $CaCl_2$, 1 mM $MgCl_2$, 50 mg/ml gentamicin, 2.5 mM Na-pyruvate and 5 mM HEPES, pH 7.6) at 18°C for 24 hr. For measuring glutamate efficacy, cells were clamped at −60 mV and perfused with $Mg^{2+}$-free extracellular solution containing (mM): 100 NaCl, 0.3 $BaCl_2$, 5 HEPES (adjusted with KOH to 7.3), 100 µM Glycine and 10 µM DTPA (zinc chelator- added before the experiment). Glutamate concentrations ranged from 0.1 to 100 µM. For Glycine efficacy of GluN1a (*wt*, E406C), extracellular solution contained 10 µM glutamate and glycine concentrations ranged from 0.1 to 10 µM. Recordings were performed with the use of a Dagan CA-1 amplifier (Dagan Corporation), controlled by the Digidata-1440 board and pClamp10 software package (Axon Instruments).

## Illumination, fast MAG photoswitching and fast glutamate uncaging

Most electrophysiological experiments were performed with illumination that was applied to the entire field of view using a Polychrome V monochromator (TILL photonics) through 20 or 40x objectives. For organotypic slice recordings (*Figure 6*) illumination was applied using a Lambda DG4 high speed wavelength switcher (Sutter instruments), with a 380 nm and a 500 nm filters through a 20x objective. Fast, millisecond photoswitching (*Figure 1f–h*) and fast photouncaging (*Figure 1—figure supplement 2g,h*) was achieved with a laser spot illumination system (*Reiner and Isacoff, 2014*). In brief, the output of a 375/488 nm dual diode laser module (Omicron LDM: 375 nm 200 mW multimode, 488 nm 80 mW single-mode) was coupled into a UV/VIS multi-mode fiber (OZ Optics, 10 µm, NA 0.1), the divergence of the exiting light reduced by threefold magnification of the fiber end, and the collimated beam directed to the objective. The laser output was controlled with TTL pulses and analog power modulation. Single spine illumination was performed on a confocal microscope (Zeiss 780-upright confocal) equipped with a 405 nm laser (100 µW at the objective). To note, we consistently used near-UV illumination (365–405 nm) for photoactivation, as nucleotides, DNA and proteins do not efficiently absorb in this region (*Forné et al., 2012*; *Schmid, 2001*), making these wavelengths and technique undamaging and compatible with biological preparations.

Targeted illumination with simultaneous electrophysiological recordings were performed on a Zeiss 780-upright confocal microscope (e.g. Figure 1i, dashed box and Figure 1—figure supplement 1b, c) .

For sculpting the photo-activation and deactivation profile of light-agonized GluN2A(V713C) the 488 nm light intensity was modulated. Typically 4–8 traces were averaged and the apparent activation and photo-deactivation kinetics were fitted with single exponential functions. For photouncaging experiments, 4-methoxy-7-nitroindolinyl-caged-L-glutamate (MNI-glutamate, Tocris) was added to the bath (0.5 mM) and uncaged with a short 375 nm laser pulse centered at the cell (~50 W/mm$^2$, ~ 15 µm spot diameters). Uncaging pulses of 0.5 ms, 1 ms and 2 ms yielded identical activation kinetics, demonstrating that these reflected the intrinsic receptor activation kinetics rather than concentration-dependent second order binding. Deactivation due to diffusion of glutamate out of the uncaging site occurred on the timescale of seconds and became slower, as more glutamate was uncaged with longer pulse lengths. Rise times (10–90%) were used to describe the speed of activation and were compared to *wt*-receptors using two-tailed, unpaired t-tests.

## Cell viability assay

Hippocampal cells (15 DIV) were assayed for cellular and membrane viability following different treatments: 1) incubation with 300 µM L-MAG1 in NMDG-labeling solution (see above), 2) with 0.3% DMSO, 3) with NMDG-labeling solution or 4) untreated. Cells were labeled using the Neurite outgrowth kit (Molecular Probes) and red and green fluorescence was acquired using a Zeiss 780-upright confocal microscope. Assessment was done by comparing multiple coverslips (N= 2–9) at same cellular density. Confocal images of red (neurite content) and green (cell viability) fluorescence were taken under identical imaging settings and identical region sizes (as described by the manufacturer). Data is presented as mean ± SEM and compared using one-way analysis of variance (ANOVA) with a *post hoc* Tukey test, all-pairwise analysis (n.s.; not significant).

## Single spine imaging of calcium and structural plasticity

Single spine imaging experiments were performed in organotypic slices obtained from GluN2A-knockout neonate mice at room temperature (~25°C) in Mg$^{2+}$-free and high-Ca$^{2+}$ACSF containing (in mM): 118.9 NaCl, 2.5 KCl, 1 NaH2PO4, 26.2 NaHCO3, 11 glucose, 4–5 CaCl$_2$, 0.001 TTX and 2.5 MNI-glutamate, oxygenated with 95% O2 and 5% CO2, as typically described elsewhere (*Matsuzaki et al., 2004*; *Okamoto et al., 2004*; *Harvey and Svoboda, 2007*; *Harvey et al., 2008*; *Makino and Malinow, 2009*). TTX and glycine were added and MK-801 removed at least 25 min before start of the imaging experiments. Images were taken using a laser scanning microscope; Zeiss 780-upright confocal microscope. Photo-uncaging of MNI-glutamate and photo-activation of MAG (300 µM) for either photo-*agonism* or photo-*antagonism* was triggered by illumination with a 405 nm laser at ~1 Hz for a total of 1–2 min, as previously described for MNI-uncaging (*Matsuzaki et al., 2004*; *Nishiyama and Yasuda, 2015*).

To test the ability of photo-agonism to induce a rise in calcium in single spines, we expressed a fusion of the calcium indicator R-GECO1.0 (*Zhao et al., 2011*) to the C-terminus of GluN2A(V713C) (GluN2A(V713C)-R-GECO) and co-transfected with additional soluble R-GECO, to improve the calcium signal detection within the entire spine head, under excitation at 561 nm, and eGFP, to enable simultaneous imaging of spine size under excitation at 488 nm. In the photo-agonized (100 µW for 100 µs/pixel) spine, the R-GECO signal decayed over tens of seconds, even though the photo-agonized GluN2A containing NMDAR remains open without decay for tens of seconds, due to bleach induced by 3 different illuminating lasers (405, 488 and 561 nm). We applied this unique combination of laser-illumination, to bypass the known artifacts of R-GECO, which undergoes unique reversible bleaching (*Shaner et al., 2008*).

Spine morphological changes were assessed by time-lapse Z-stack images of single spines collected using a 20x/NA=1.0 water immersion objective (digital zoom: 10–13) by imaging space-filling cytosolic tdTomato with a 561 nm laser. 3-dimensional projection images (of 512x512 or 1024x1024 pixels) were exported (Zen 2011 software, Zeiss) for analysis with a custom MATLAB program (*Peled and Isacoff, 2011*). We measured the fluorescence of the spine (as shown in [*Zhang et al., 2008*]), which is proportional to spine-head volume (*Holtmaat et al., 2005*), and normalized it to the fluorescence of the shaft to correct for any changes that may have occurred to fluorescence; resulting from bleach or movement (*Nimchinsky et al., 2004*). Nearby spines typically consisted of spines located on the same dendrite (<10 µm from photo-stimulated spine) found in the same field of view as the photostimulated spine or atop dendrites that ran across the field of view, nearby the photostimulated region.

To determine whether the ability to induce spine expansion was restored to CA1 neuronal spines, in slices prepared from GluN2A-knockout mice, neurons were transfected with the GluN2A(V713C) subunit and single spines were assessed for expansion by uncaging MNI-glutamate. MNI-glutamate (2.5 mM final concentration) was allowed to diffuse into the slice during 1–2 min of superfusion with Mg$^{2+}$-free bathing solution, in the presence of 2 µM TTX, before photo-uncaging. Photo-uncaging with 405 nm light (100 µW for 100 µs/pixel) at a spot of the same lateral size as the spine head, but located 0.5 µm away (*Figure 8—figure supplement 1b*). This illumination reliably induced expansion of the spine below the spot of illumination (*Figure 8—figure supplement 1b*, dashed circle) (*Hardingham and Bading, 2010*; *Lee et al., 2009*), indicating rescue of structural plasticity by expression of the cysteine-bearing subunits.

## Generation of GluN2A(G712C) transgenic zebrafish

To generate a stable transgenic line of LiGluN2A zebrafish, we inserted the rat-GluN2A(G712C) gene with a N-terminal GFP tag under control of a 10X-UAS sequence into a pT2KXIG-delta-IN vector containing Tol2 transposon recognition sites (*Suster et al., 2009*) using *EcoR*I and *Not*I restriction sites. The resulting pT2-LiGluN2A(G712C) construct was diluted to 50 ng/µL with 0.25 ng Tol2 transposase and 0.05% Phenol Red (Sigma). Injected embryos were raised to sexual maturity and outcrossed to identify founder (F0) fish. F1 fish were genotyped for GluN2A(G712C) using primers 5'-TGCA GAG AAT CGG ACC CAC T-3' and 5'-TCG ATC ACT GCC CTC ACT GT-3'. F1 *Tg[UAS: GluN2A(G712C)]* fish were outcrossed to a pan-neuronal transgenic Gal4 line, *Tg(s1101t-gal4)*. Double transgenic *Tg[s1101t- gal4;UAS-GluN2A(G712C)]*; fish grew to sexual maturity with no obvious phenotypes.

To determine whether exposure to MAG affected the health or development of zebrafish larvae, we assayed basic spontaneous and evoked behaviors of freely swimming animals for up to 2 days post-treatment. 5 dpf wildtype larvae were bathed in either with 150 µM MAG or vehicle only under the same conditions as used in the LiGluN2A experiments. Fish were transferred to 48-well plates (1 fish per well with 800 µL fresh E3) and placed in an imaging arena (*Levitz et al., 2013*). After a 10 min acclimation period, spontaneous behavior was filmed for 10 min. Movies were analyzed, using custom made Matlab programs, to determine population statistics for metrics representative of spontaneous motor behaviors: distance traveled, time spent resting, place preference, and the frequency and duration of spontaneous swim bouts. MAG-treated animals exhibited no significant difference in any measure compared with DMSO-treated controls.

## Time lapse imaging of retinal ganglion cell axonal arbor development

For visualization of individual retinal ganglion cells, embryos from incrosses of *Tg[s1101t-gal4;; UAS-GluN2A(G712C)]* fish were injected with *pou4f3-gal4:UAS-mGFP* DNA (mGFP; monomeric membrane-bound due to palmitoylation sequence from gap43). Injected embryos were reared in E3 containing 0.005% 1-phenyl 2-thiourea (PTU) and screened based on GFP expression at 48 hpf. Typically, injected embryos had 1–3 non-overlapping labeled axons. At 120 hpf, *pou4f3- gal4:UAS-mGFP* positive fish were assessed for spontaneous swimming and inflated swim bladder as markers for normal development. Healthy animals were divided into two groups to receive treatment in either 150 µM MAG-0 in E3 with 0.3% DMSO or in E3 with 0.3% DMSO for 40 min at 28.6°C. All fish then were rinsed twice in fresh E3 and allowed to recover for 2 hr at 28.6°C.

Fish were then embedded in 1.4% low-melting agarose in E3 for an initial imaging session to measure baseline morphology of the RGC projections. Following the baseline imaging session, each animal was freed from agarose and transferred to fresh E3 containing 0.005% PTU to recover from anesthesia. Fish typically recovered spontaneous swimming within 1 hr. Between 128 hpf and 168 hpf, fish were subjected to a photo-antagonism protocol described below. Each fish was then remounted in agarose for a second imaging session to assess RGC growth. Following the second imaging session, fish were euthanized in 0.4% tricaine. Genomic DNA was then extracted from individual whole animals in 60 µl volume (*Meeker et al., 2007*) and genotyped for the *UAS-GluN2A (G712C)* transgene using the PCR primers described above.

At 128 hpf, each fish was transferred to a 48 well plate in 800 µl E3 with 0.005% PTU and placed in a custom imaging chamber equipped with an array of 405 nm LEDs centered under each well. The LED array was controlled by a DAQ (National Instruments) programmed to produce 10 s flashes every 30 min. The imaging chamber maintained ambient lighting on a 14 hr light/10 hr dark cycle with an ambient temperature of ~26°C.

Twenty minutes prior to each imaging session, each fish was mounted dorsal-side up in a glass-bottom dish in 1.4% low-melting-point agarose dissolved in E3, and anesthetized with 0.02% tricaine. Z-stacks with 0.35–0.5 µm steps were obtained using a 2-photon Zeiss LSM710 microscope equipped with a 3 W Ti-Sapphire laser (Coherent, Chameleon Ultra) tuned to 920 nm. Minimal laser intensity was used (<20 mW measured at the front of the objective). A long working distance water-dipping 40X/1.0 NA objective was used to acquire all images. In each 3- dimensional image, RGC axon arbors were traced using the Simple Neurite Tracer plug-in for ImageJ (*Longair et al., 2011*). Sholl analysis was performed on each isolated 3-dimension arbor, where spheres were centered on the first branch point remaining at 7 dpf (*Figure 9c*, inset).

## Availability of MAG

MAG photo-switches are now available commercially from Aspira Scientific: http://www.aspirasci.com/neuroscience-probes

## Animal usage

All animal experiments were done under oversight by the University of California institutional review board (Animal Care and Use committee). This study was performed in strict accordance with the recommendations in the Guide for the Care and Use of Laboratory Animals of the National Institutes of Health. All of the animals were handled according to approved institutional animal care and use committee (IACUC) protocols (AUP-2015-04-7437) of the University of California. Every effort was made to minimize suffering.

## Statistical analysis

Results are shown as mean ± SEM. Data sets of >3 points of data were tested for Normality (normal distribution), followed by a multiple group comparison, using one-way analysis of variance (ANOVA) with a *post hoc* Tukey test. Data sets which failed the normality test were followed by ANOVA on ranks (Mann-Whitney Rank Sum Test). All-pairwise analysis and two group comparisons were done using two-tailed, T-test or paired T-test, when applicable (SigmaPlot™, 2011, Systat Software Inc., San Jose, CA). Rank Sum test was applied whenever normality failed for two group comparison. Asterisks indicate statistically significant differences as follows: *p<0.05; **p<0.01; ***p<0.001, n.s.,

not significant. Box plots display medians, 10th and 90th percentiles and outliers (filled circles). Individual data points, when displayed, are found to the right of the box plot as filled squares.

## Acknowledgements

We thank Noam Ziv for NMDA receptor cDNAs, Masayoshi Mishina and David Lovinger for GluN2A-knockout mice, Holly Aaron for help with microscopy, Marta Soden for guidance on hippocampal slice culture and Zhu Fu and Matthew Truong for help with molecular biology and cell culture. We also thank Mu-Ming Poo, Richard Kramer, Dan Feldman, Doris Fortin and Matthew Volgraf for helpful discussion. We thank Joshua Levitz for helpful discussion and help with autapse recordings. Support was provided by the National Institutes of Health Nanomedicine Development Center for the Optical Control of Biological Function (NIH 2PN2EY018241) (DT and EYI) and U01 NS090527 (EYI), National Science Foundation (NSF-1041078) (EYI), HFSP (RGP0013/2010) (EYI), ERC Advanced Grant (DT), National Science Foundation predoctoral fellowship (SS), German Research Foundation post-doctoral fellowship (DFG RE 3101/1-1) (AR), and CNRS (INSB) (AG).

## Additional information

### Funding

| Funder | Grant reference number | Author |
|---|---|---|
| National Science Foundation | Predoctoral fellowship | Stephanie Szobota |
| Deutsche Forschungsgemeinschaft | RE 3101/1-1 | Andreas Reiner |
| Centre National de la Recherche Scientifique | INSB | Alice Guyon |
| National Institutes of Health | 2PN2EY018241 | Ehud Y Isacoff<br>Dirk Trauner |
| European Research Council | Advanced Grant | Dirk Trauner |
| National Institutes of Health | U01 NS090527 | Ehud Y Isacoff |
| National Science Foundation | NSF-1041078 | Ehud Y Isacoff |
| Human Frontier Science Program | RGP0013/2010 | Ehud Y Isacoff |

The funders had no role in study design, data collection and interpretation, or the decision to submit the work for publication.

### Author contributions

SB, SS, AR, ECC, AG, Conception and design, Acquisition of data, Analysis and interpretation of data, Drafting or revising the article; MAK, Drafting or revising the article, Contributed unpublished essential data or reagents; TX, Conception and design, Analysis and interpretation of data; DT, Conception and design, Contributed unpublished essential data or reagents; EYI, Conception and design, Analysis and interpretation of data, Drafting or revising the article

### Author ORCIDs

Ehud Y Isacoff, http://orcid.org/0000-0003-4775-9359

### Ethics

Animal experimentation: This study was performed in strict accordance with the recommendations in the Guide for the Care and Use of Laboratory Animals of the National Institutes of Health. All of the animals were handled according to approved institutional animal care and use committee (IACUC) protocols (AUP-2015-04-7437) of the University of California. Every effort was made to minimize suffering.

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
