## [Decision Letter]

Thank you for submitting your work entitled "A family of photoswitchable NMDA receptors" for consideration by *eLife*. Your article has been reviewed by two peer reviewers, and the evaluation has been overseen by a Reviewing Editor and Gary Westbrook as the Senior Editor.

The reviewers have discussed the reviews with one another and the Reviewing Editor has drafted this decision to help you prepare a revised submission.

Summary:

The reviewers were favorably impressed with the main achievement of the study, i.e., the development of NMDA receptor subunits with photoswitchable tethered ligands, whose introduction into cells allows NMDA receptors of specific compositions to be turned on and off by light. These optically gated subunits set the stage for a wide range of studies of NMDA receptor contributions to a variety of neural processes. (The Reviewers' specific comments are included in "General Assessment" below.)

Essential Revisions:

The reviewers brought up several points for revision. Since the primary contribution of the work is to generate a tool, its experimental utility must be more explicitly demonstrated, and its shortcomings and/or limitations must be clearly explained. As one reviewer said in the discussion, "At the moment, the excellent data are slightly diminished by glossing over important details." The points to address therefore center on showing and/or clarifying how broadly applicable the technique in its current form actually is. They are summarized here, and more details are given in the in "Essential Revisions" of the Reviewers' comments below.

1) Please provide better quantification of the effect of the photoswitchable receptors, especially regarding how the effect of photostimulation varies with efficiency of labeling and the ratio of modified to wild-type receptors. Comparison of existing results (in wild-type autapses and KO slices) to data from knockout autapses and/or with overexpression in wild-type slices would be helpful.

2) Because the main power of the technique will likely come from applications in preparations that are more intact than autapses and organotypic slices from neonates (e.g., acute slices and/or more intact preparations), as well as from subcellular control of receptors, please show or assess how feasible it will be to (de)activate these receptors with a technique like 2-photon absorption.

3) Since use of these receptors depends on application of the MAG ligands, please provide an indication of or instructions regarding their availability.

4) Some of the experiments appear statistically underpowered. Please reassess the statistical methods and revise accordingly (either by obtaining additional measures or modifying the analysis and/or conclusions).

5) In general, please ensure that the claims of the manuscript are commensurate with the data.

Reviewers' Comments:

General assessment comments:

This paper is an example of the mature stage of the authors' fascinating project of photo pharmacology with MAG and related compounds that started about 10 years ago. The molecular engineering and characterisation of the derivatized receptors is very impressive. It's satisfying that the authors seem to be able to bend the receptors to their will. The scope of the work is broad and it fits well to *eLife*.

This exceptional study describes the development of NMDAR subunits that can be modified with light to allow spatiotemporal control over NMDAR signaling. The authors employ a previously developed strategy to render different NMDAR subunits sensitive to light, by incorporating cysteines in key locations in the glutamate binding pocket of GluN subunits, allowing attachment of photoswitch-tethered ligands (a maleimide and a photoisomerizable azobenzine-glutamate) that when exposed to UV light enable activation or inactivation. The authors performed a detailed analysis of photo-antagonism and photo-agonism of these constructs in HEK cells, cultured neurons and in brain slices, demonstrating their ability to modulate NMDAR signaling at synapses to control excitation and plasticity with unprecedented precision. One major advantage of this approach is that the alteration of receptors is reversible, as the ligands can be switched from inactive trans to active cis states by different wavelengths of light. Remarkably, overexpression of these modified receptors does not appear to alter normal synaptic function (e.g. lead to enhanced NMDARs at synapses or alter the ability of endogenous glutamate to activate these receptors) and activation of photoswitches appears affected primarily NMDARs, despite the many accessible cysteines found in other proteins. Thus, it is likely that these tools will be very useful in defining the roles of distinct NMDAR subunits in synaptic function and plasticity, and provide a new optogenetic approach for eliciting cellular excitation.

Essential Revisions (expanded version):

1) The experiments in slices, which potentially provide the real excitement here, are somewhat patchy. Some are spectacular, e.g. Figure 8 is very strong. In other aspects, the approach is piecemeal. Effects on spine size are of exceptional interest, but what does photo-antagonism of synaptic currents look like on a KO? Is the effect limited by labelling in slice? Or by the ratio of wt:mut subunits? It's possible that long stimulation overcomes this problem for the photo-agonism experiments. No matter the results on LTP and spines, which are strong, the labelling/efficiency question is very important. In Figure 5, the effect is small and not much of much practical use. What happens on a KO background? Is the effect small because of limited labelling, or because of limited expression, or both or is it something else? This paper primarily introduces a tool. If the authors want their tool to be widely accepted, this kind of information is essential. The simplest way to address this is – to what extent are synaptic currents inhibited on KO background? If this experiment can't or won't be done, then there must be an explanation of this caveat in the paper. I would not include the latter (albeit impressive) effects on spines as a justification that the effect of light is all or nothing. We don't know how much NMDA current is needed for the effects on spines.

2) The authors have demonstrated the efficiency of modification using visible light. However, in intact preparations it is often desirable to manipulate signaling in subcellular domains. Although visible light is useful for dissociated cells and cells near the surface of cultured slices, scattering limits the utility of this approach when cells are deeper in tissue. It would significantly enhance the study if the authors were to provide an evaluation of the ability of these photoswitches to be modified by two photon absorbance.

3) The primary impact of the study lies in the ability of these variants and this strategy to be employed in a variety of contexts. In this regard, it is important to recognize that the manipulations require L-MAG0 and L-MAG1, in addition to the modified receptors. Thus, the authors should provide some information in the paper about the availability of the photoswitch ligands.

4) For statistical analyses, t-tests or one-way ANOVA with posthoc Tukey tests were performed. These tests require that the data are normally distributed. However, the authors do not indicate whether tests of normality were performed, and in places where individual datapoints are provided (e.g. Figure.4B-D), they do not appear to be normally distributed. There is some concern that for experiments such as those shown in Figure 5 that the analysis is underpowered. If variance is high and multiple samples are compared, it is relatively easy to find no significance difference between groups. If the authors were to address these issues, it would increase confidence in the comparisons.

[Editors' note: further revisions were requested prior to acceptance, as described below.]

Thank you for resubmitting your work entitled "A family of photoswitchable NMDA receptors" for further consideration at *eLife*. Your revised article has been favorably evaluated by Gary Westbrook (Senior editor), a Reviewing editor, and two reviewers. The manuscript has been greatly improved. There are a few remaining issues that must be addressed before acceptance, as outlined below:

The reviewers appreciated the extent and quality of the revisions. The main question that arose had to do with the reporting of results relating to the efficiency of the manipulations in a manner that would be most useful the readers who might wish to implement the technique. A specific issue was the separation of responding and non-responding wild-type cells after transfection. The basis for this separation was not clearly stated in the manuscript, and Figure 5 came across as ambiguous, since the 4 non-responding cells of 10 tested were neither reported nor illustrated in the main text. Because the strength of this manuscript lies in the development of tools, the reviewers agree that a clear statement here and throughout the manuscript of n values and fraction of cells responding is crucial for others to evaluate the usefulness of the technique. A related point is that in our discussion of the reviews, the reviewers raised the possibility that the fraction of non-responders could result either from efficiency of labeling (as stated in the text) or from expression of the target (not stated in the text), which you may wish to comment upon. The full reviews are included below.

*Reviewer #1:*

The authors have made a comprehensive response that largely answers all the previously raised points of difficulty.

However, the analysis that has been added to answer concerns about the fundamental mechanisms underlying partial effects is itself possibly problematic.

Figure 5—figure supplement 1 indicates 4 non-responding cells. My interpretation is that these cells were transfected, exposed to MAG but did not get inhibited. Otherwise, they would be controls of some description. 6 further cells were used (in each case pooled across 2 conditions) and are shown in the main figure. The non-responders are not included in the average effect. They did have typical synaptic currents, in 3 out of 4 cases. The statistical analysis for inhibition included in the supplementary figure seems distinct from that for the tau and epsc size. Overall, for whatever reason, be it weak/variable expression of the target subunits, or poor labelling with the optically active element, the impression I have here is that 6/10 experiments gave inhibition, but 4/10 did not. Thus, the range cited in the third paragraph of the Discussion is misleading. If the authors want to exclude these cells for an objective reason, they should state in the main figure, and where they cite the range, that 4 cells were excluded.

Likewise, there are quite a few cells in the supplement to Figure 3 that might be thought of as non-responders.

These results are described by the authors in their response and in the discussion as "satisfyingly reliable". If I have understood correctly, these data indicate that, on a wild-type background, the method is not that reliable. There is a stark contrast here with the very reproducible and important effects when rescuing GluN2A expression on a knockout background, and the new results in zebrafish which are also convincing.

There is no sense in which this limitation with wild-type backgrounds detracts from the overall study and its groundbreaking nature. The weak and variable effects are not surprising and can be circumvented with standard methods (genetically-targeted animals). But at the moment, I have the impression that authors are not accurately reporting the work on wild-type backgrounds.

Further, are there any other panels where non-responding cells are excluded and, if so, on what basis?

*Reviewer #1 (Additional data files and statistical comments):*

The statistical analysis for inhibition included in Figure 5—figure supplement 1 seems distinct from that for the tau and epsc size. Why?

*Reviewer #2:*

The authors have done a nice job of responding to the previous concerns, in particular by extending the statistical analyses, providing additional experimental details, softening some of their earlier conclusions, and providing additional discussion. They have also included additional results showing that GluN2 photoantagonism in vivo in zebrafish induces axonal refinement deficits that are comparable to systemic MK801 exposure, providing another demonstration of utility. The authors have not provided data regarding the efficiency with which these photoswitches can be induced using two photon absorption, but they indicate that new MAG photoswitches will be developed based on strategies already shown to work with kainate and metabotropic receptors. It would have been nice to include data on the existing molecules to guide other investigators with their use, but I don't see this as an essential revision.

---

## [Author Response]

*Essential Revisions:*

*1) Please provide better quantification of the effect of the photoswitchable receptors, especially regarding how the effect of photostimulation varies with efficiency of labeling and the ratio of modified to wild-type receptors. Comparison of existing results (in wild-type autapses and KO slices) to data from knockout autapses and/or with overexpression in wild-type slices would be helpful.*

We have addressed the two possible sources of variability between experiments: A) expression level of the cysteine-substituted (photoswitch-ready) LiGluN subunit and B) level of labeling with the MAG photoswitch. In addition, to put the degree of efficacy and variability in context of conventional pharmacology, we have added C) a new experiment that compares MAG photoswitching of LiGluN-containing NMDA receptors in transgenic animals to conventional pharmacological manipulation of native NMDA receptors in control animals.

We cannot compare between KO data and autaptic data because they were obtained from different species: autaptic currents and all other cultured cell data were from C57BL-*wt* rats, whereas the KO data is was from GluN2A-KO mice (Sakimura, Kutsuwada et al. 1995).

A) To assess expression levels of the LiGluN subunits we further investigated the relationship between the size of the photo-response (in the case of photo-agonism the amplitude of the photo-current; in the case of photo-antagonism the amplitude of the current reduction by light) and the total current induced by agonist or synaptic glutamate release, now shown in two new panels (Figure 3—figure supplement 1 and Figure 5—figure supplement 1). We observe a moderate correlation between the photo-responses and total NMDA-induced current (R=0.709, P<0.0001, n=22; Figure 3—figure supplement 1) and no correlation for the autaptic recordings (R= 0.543, P=0.297, n=6; Figure 5—figure supplement 1). This is consistent with our initial conclusion that expression of the engineered subunits does not induce a substantial overexpression of NMDA receptors (as assessed in Figure 1—figure supplement 3 and Figure 5).

B) We have no direct way of measuring MAG conjugation efficiency (MAG is not fluorescent), but given (A), above, and the general experience in the field of variable penetration of small organic molecules of the size of MAG (such as fluorescent dyes) into synapses in dissociated primary neuron cultures and in brain slice, we conjecture that variability in photoswitch efficacy is primarily due to variability in MAG conjugation efficiency.

In accordance with the reviewer request, we now provide data from individual recordings to key figure panels to show the experiment-to-experiment variability (see revised Figure 2, Figure 2—figure supplement 2, Figure 3—figure supplement 1 and Figure 5). The raw data for individual autapses was already provided in Figure 5 (open circles), but are now also included in a new panel (Figure 5—figure supplement 1) with additional non responding cells as comparison, for easy assessment of the variability in the experiment.

C) We have also added a new section (including two Figures) to compare the variability of our optogenetic photo-pharmacological block of NMDA receptors using LiGluN with the variability seen with conventional pharmacology for blocking NMDA receptors. We did this in the challenging biological context of the intact zebrafish brain in vivo. We compared the effect of photo-antagonism of NMDA receptors with LiGluN2A(G712C) versus the activity-dependent pore blocker MK-801 on the activity-dependent development of retinal ganglion cell axonal arbors in the optic tectum. We find that the manipulations have both a similar magnitude of effect and a strikingly similar degree of variability (new Figure 9 and its supplement).

*2) Because the main power of the technique will likely come from applications in preparations that are more intact than autapses and organotypic slices from neonates (e.g., acute slices and/or more intact preparations), as well as from subcellular control of receptors, please show or assess how feasible it will be to (de)activate these receptors with a technique like 2-photon absorption.*

We agree that 2-photon photoswitching would be very valuable to achieve good 3D light confinement and better tissue penetration in acute brain slice and in vivo. We and others have recently designed three new classes of MAGs (Kienzler, Reiner et al. 2013; Gascon-Moya, Pejoan et al. 2015) that we, and others, show to work well with 2-photon (2P) photoactivation of a related iGluR (kainate receptor) and an mGluR (Izquierdo-Serra, Gascon-Moya et al. 2014; Carroll, Berlin et al. 2015; Gascon-Moya, Pejoan et al. 2015). This approach will be adapted to LiGluNs as well. This is now mentioned in the text (Discussion, seventh paragraph).

In preparations such as *C. elegans, Drosophila* and zebrafish, in which precise genetic targeting can be readily achieved (i.e. of the photoswitch-ready subunit) and where the nervous system is accessible to light and much less scattering (especially in the zebrafish), it is already possible to accomplish MAG photo-control of glutamate receptors with conventional 1-photon (1P) illumination, as we showed previously in zebrafish sensory neurons (Szobota, Gorostiza et al. 2007), in the central pattern generator circuit for swimming (Wyart, Del Bene et al. 2009; Janovjak, Szobota et al. 2010) and pan-neuronally (Levitz, Pantoja et al. 2013). We now add a new section to the paper (final section of Results) that demonstrates effective optical control with 1P light in vivoin transgenic zebrafish expressing the LiGluN2A(G712C) subunit (new Figure 9 and Figure 9—figure supplement 1).

*3) Since use of these receptors depends on application of the MAG ligands, please provide an indication of or instructions regarding their availability.*

MAG photo-switches are now available commercially from Aspira Scientific: http://www.aspirasci.com/neuroscience-probes (noted in subsection “Availability of MAG”).

*4) Some of the experiments appear statistically underpowered. Please reassess the statistical methods and revise accordingly (either by obtaining additional measures or modifying the analysis and/or conclusions).*

We reassessed our statistical analysis and figure presentations and now show the complete data sets and statistical analysis. We replotted panels where the data did not distribute normally as Box plots and included results for normality tests and ANOVA on ranks that we performed on the data (SigmaPlot 2011; V11.0 in Figure 1—figure supplement 3, right panel, Figure 2 and corresponding supplement 2, Figure 4—figure supplement 1 with their corresponding supplement 2, Figure 5 and corresponding supplement 1, Figure 9 and corresponding supplement 1C). We have also expanded our description of the statistical analysis in Methods (subsection “Statistical analysis”). Importantly, our interpretation of the results did not change based on these new statistical tests. We have, however, moderated our description in the Discussion, as suggested by the reviewers.

*5) In general, please ensure that the claims of the manuscript are commensurate with the data.*

We have followed this suggestion, as documented below.

*Essential Revisions (expanded version):*

*1) The experiments in slices, which potentially provide the real excitement here, are somewhat patchy. Some are spectacular, e.g. Figure 8 is very strong. In other aspects, the approach is piecemeal. Effects on spine size are of exceptional interest, but what does photo-antagonism of synaptic currents look like on a KO? Is the effect limited by labelling in slice? Or by the ratio of wt:mut subunits? It's possible that long stimulation overcomes this problem for the photo-agonism experiments. No matter the results on LTP and spines, which are strong, the labelling/efficiency question is very important. In Figure 5, the effect is small and not much of much practical use. What happens on a KO background? Is the effect small because of limited labelling, or because of limited expression, or both or is it something else? This paper primarily introduces a tool. If the authors want their tool to be widely accepted, this kind of information is essential. The simplest way to address this is* –

*to what extent are synaptic currents inhibited on KO background? If this experiment can't or won't be done, then there must be an explanation of this caveat in the paper. I would not include the latter (albeit impressive) effects on spines as a justification that the effect of light is all or nothing. We don't know how much NMDA current is needed for the effects on spines.*

As described in the response to the Essential Revisions Summary point 1, the degree of photo-activation or block is influenced by: i) the fraction of cysteine-substituted ("photoswitch ready") subunits that are incorporated into the GluN2 "slot" of NMDA receptors and ii) the efficiency of labeling those subunits with the photoswitch. In the case of the KO, (i) is optimized due to absence of the native GluN2A. However, even in neurons isolated from WT animals expressing the WT GluN2A (as shown in Figure 3 and summary in F, and in new Figure 3—figure supplement 1), photo-antagonism can be remarkably potent (>50%; i.e. equivalent to the efficacy obtained with classical blockers, such as the GluN2A antagonist NVP-AAM077, when applied at concentrations that ensure selectivity for GluN2A-containing receptors).

The degree of photo-block of NMDA receptor EPSCs in the autapses from WT rat neurons (Figure 5) is indeed small on average (mean 25-35%, range ~20-50%). We elected to test this in the WT context because this represents the strongest test of the efficiency of the approach for generating NMDARs that are light-controlled in a background where the LiGluN subunit competes with WT subunits to assemble into receptors that are properly trafficked into synapses. Despite the smaller fractional optical control, it is satisfying that the control is reliable. We have added into the Results, Methods and Figure Legends an explicit mention for each experiment of whether it was performed in the GluN2A-KO or in WT (e.g.” Cultured hippocampal neurons from wt rats (C57BL) transfected with the photo-antagonizing 261 GluN2A(G712C) subunit in combination with photo-antagonizing GluN1a(E406C),”; in *all* relevant figure legends and Methods, subsections “Cell culture and transfection” and “Organotypic hippocampal slice culture preparation and biolistic transfection”). We provide a caveat that explains that the degree of photo-block in organotypic slices from the GluN2A-KO that were used for the LTP and spine expansion experiments is likely significantly greater than the small degree of photo-block seen in WT autapses where only a fraction of the NMDA receptors are under optical control (in the Discussion, third paragraph).

We have added several additional panels to show the variability of photoswitching in several different preparations and experimental paradigms to the figures that did not already include such data (e.g. Figure 2, Figure 5, Figure 9).

In NR2A-KO cells, where we could block the remaining NR2B-containing receptors using ifenprodil for example, we would still be limited by variability in MAG labeling, much like the results obtained from HEK-cell recordings, that lack native NMDA receptors and so necessarily contain two subunits of LiGluN2A in each and every receptor found at their membrane but which, nevertheless show the same variability in photo-current amplitude compared to agonist-evoked current.

We do not know how much NMDA receptor current is needed to induce spine expansion, but this uncertainty applies as well to the “classical” method of studying spine morphological dynamics of photo-uncaging of caged glutamate (MNI-glutamate). In fact, most uncaging experiments are performed in high external Ca^2+^ (i.e. >2.5 mM)(e.g. (Matsuzaki, Honkura et al. 2004; Okamoto, Nagai et al. 2004; Harvey and Svoboda 2007; Harvey, Yasuda et al. 2008; Makino and Malinow 2009); as described in the first paragraph of the subsection “Single spine imaging of calcium and structural plasticity“, taking into consideration the variability in the number of responsive spines and threshold for activation (Lynch, Kramar et al. 2013), the fact that synaptic transmission nor uncaging saturates synaptic NMDA receptors (Mainen, Malinow et al. 1999) and because the number of receptors per spine is highly variable.

Our ability to induce or block spine expansion (Figure 7 and Figure 8) represents a fundamental practical success since this cellular change is thought to be key to LTP (Matsuzaki, Ellis-Davies et al. 2001; Matsuzaki, Honkura et al. 2004; Zhang, Holbro et al. 2008), which we rephrased in the text (subsection “Single spine imaging of calcium and structural plasticity”). The proof of the relevance of this to the study of the mechanism of long-term synaptic plasticity and its role in circuit function is obtained by using an electrophysiological assay to confirm that photo-antagonism blocks LTP induction as measured electrophysiologically via changes in EPSC amplitude (Figure 6).

*2) The authors have demonstrated the efficiency of modification using visible light. However, in intact preparations it is often desirable to manipulate signaling in subcellular domains. Although visible light is useful for dissociated cells and cells near the surface of cultured slices, scattering limits the utility of this approach when cells are deeper in tissue. It would significantly enhance the study if the authors were to provide an evaluation of the ability of these photoswitches to be modified by two photon absorbance.*

As described in the response to the Essential Revisions Summary point 2, we and others have generated three new variant versions of MAG (Kienzler et al., 2013; Gascon-Moya, Pejoan et al. 2015) that work well with 2-photon (2P) photoactivation of a related iGluR (kainate receptor) and an mGluR (Carroll, Berlin et al. 2015; Gascon-Moya, Pejoan et al. 2015). This approach can next be adapted to LiGluNs. This is now mentioned in the text (Discussion, seventh paragraph).

In addition, in preparations such as *C. elegans, Drosophila* and zebrafish, where precise genetic targeting can be readily achieved (i.e. of the photoswitch-ready subunit) and where the nervous system is accessible to light and (especially in the zebrafish) much less scattering, it is already possible with conventional 1-photon (1P) illumination to accomplish MAG photo-control of glutamate receptors, as we showed previously in zebrafish sensory neurons (Szobota et al., 2007), the spinal central pattern generator circuit for swimming (Wyart et al., 2009; Janovjak et al., 2010) and pan-neuronally (Levitz et al., 2013).

We have added a new section to the paper (LiGluN2A photo-antagonism disrupts pruning […] in vivo) that demonstrates effective optical control with 1P light in vivo in transgenic zebrafish expressing the LiGluN2A(G712C) subunit (new Figure 9 and Figure 9—figure supplement 1).

*3) The primary impact of the study lies in the ability of these variants and this strategy to be employed in a variety of contexts. In this regard, it is important to recognize that the manipulations require L-MAG0 and L-MAG1, in addition to the modified receptors. Thus, the authors should provide some information in the paper about the availability of the photoswitch ligands.*

MAG photoswitches are now available commercially from Aspira Scientific: http://www.aspirasci.com/neuroscience-probes, now mentioned in Methods (subsection “Availability of MAG”).

4) For statistical analyses, t-tests or one-way ANOVA with posthoc Tukey tests were performed. These tests require that the data are normally distributed. However, the authors do not indicate whether tests of normality were performed, and in places where individual datapoints are provided (e.g. Figure 4), they do not appear to be normally distributed. There is some concern that for experiments such as those shown in Figure 5 that the analysis is underpowered. If variance is high and multiple samples are compared, it is relatively easy to find no significance difference between groups. If the authors were to address these issues, it would increase confidence in the comparisons.

As described in the response to the Essential Revisions Summary point 4, we reassessed our statistical analysis and figure presentations and now show complete data sets and analysis. This includes normality tests and ANOVA on ranks for data that do not normally distribute. Accordingly, we replotted panels to better describe the data. Importantly, our interpretation of the results did not change based on these new statistical tests. This is also noted in Methods; statistics section (subsection “Statistical analysis”) and in additional panels and supplementary figures.

[Editors' note: further revisions were requested prior to acceptance, as described below.]

*The reviewers appreciated the extent and quality of the revisions. The main question that arose had to do with the reporting of results relating to the efficiency of the manipulations in a manner that would be most useful the readers who might wish to implement the technique. A specific issue was the separation of responding and non-responding wild-type cells after transfection. The basis for this separation was not clearly stated in the manuscript, and Figure 5 came across as ambiguous, since the 4 non-responding cells of 10 tested were neither reported nor illustrated in the main text. Because the strength of this manuscript lies in the development of tools, the reviewers agree that a clear statement here and throughout the manuscript of n values and fraction of cells responding is crucial for others to evaluate the usefulness of the technique. A related point is that in our discussion of the reviews, the reviewers raised the possibility that the fraction of non-responders could result either from efficiency of labeling (as stated in the text) or from expression of the target (not stated in the text), which you may wish to comment upon. The full reviews are included below. Reviewer #1: The authors have made a comprehensive response that largely answers all the previously raised points of difficulty. However, the analysis that has been added to answer concerns about the fundamental mechanisms underlying partial effects is itself possibly problematic. Figure 5—figure supplement 1 indicates 4 non-responding cells. My interpretation is that these cells were transfected, exposed to MAG but did not get inhibited. Otherwise, they would be controls of some description. 6 further cells were used (in each case pooled across 2 conditions) and are shown in the main figure. The non-responders are not included in the average effect. They did have typical synaptic currents, in 3 out of 4 cases. The statistical analysis for inhibition included in the supplementary figure seems distinct from that for the tau and epsc size. Overall, for whatever reason, be it weak/variable expression of the target subunits, or poor labelling with the optically active element, the impression I have here is that 6/10 experiments gave inhibition, but 4/10 did not. Thus, the range cited in the third paragraph of the Discussion is misleading. If the authors want to exclude these cells for an objective reason, they should state in the main figure, and where they cite the range, that 4 cells were excluded.*

Though the 4 excluded cells were initially mentioned in the legend of Figure 5—figure supplement 1, we now added mention to the main text (Discussion, third paragraph), and we added a description of the criteria for exclusion in the legend. We note that non-responding cells were excluded from the averages shown in the main figure in both legends (main Figure 5 and Figure 5—figure supplement 1) because of the very small degree of block (<10%). These small effects (though likely real) are hard to discern from the naturally occurring rundown that results from the depletion of the ready-releasable pool of vesicles, as shown by Goda and Stevens (Goda & Stevens, 1998) (Figure 5 legend). Therefore, we applied this rigorous exclusion criteria to prevent inclusion of rundown-dependent decreases in EPSC amplitude.

We retested all of the statistics and find the statistical tests to be appropriate. eEPSC_NMDA_ size and τ_deac_ are not normally distributed for the entire data set (although 2/3 of the subgroups did have a normal distribution). As a result, these were further analyzed using Kruskal-Wallis ANOVA on Ranks. The Normality test for “% inhibition” did pass for the entire data set so that it could be tested by ANOVA for significance (Sigmaplot 10.0).

*Likewise, there are quite a few cells in the supplement to Figure 3 that might be thought of as non-responders.*

Non-transfected cells do not exhibit any observable response to light (Figure 1—figure supplement 1), thus all photo-agonistic induction of current and all photo-block of NMDA-evoked current are due to photoswitching of the LiGluNs. While some of the responses are small, they are all non-zero and thereby included (Figure 3—figure supplement 1).

*These results are described by the authors in their response and in the discussion as "satisfyingly reliable". If I have understood correctly, these data indicate that, on a wild-type background, the method is not that reliable. There is a stark contrast here with the very reproducible and important effects when rescuing GluN2A expression on a knockout background, and the new results in zebrafish which are also convincing. There is no sense in which this limitation with wild-type backgrounds detracts from the overall study and its groundbreaking nature. The weak and variable effects are not surprising and can be circumvented with standard methods (genetically-targeted animals). But at the moment, I have the impression that authors are not accurately reporting the work on wild-type backgrounds.* We have rephrased our conclusions to clarify this issue. We note that the expression of the light-gated variants in a wild-type background yielded variable effects, ranging from very strong block to no block at all (Discussion, third paragraph). These observations may result from a combination of factors, among which the variable replacement of the endogenous receptors likely plays a major role. However, in responding cells, despite the small fractional optical control, the control is reproducible (Figure 3—figure supplement 1 and Figure 5—figure supplement 1).

*Further, are there any other panels where non-responding cells are excluded and, if so, on what basis?*

No.

*Reviewer #2: The authors have done a nice job of responding to the previous concerns, in particular by extending the statistical analyses, providing additional experimental details, softening some of their earlier conclusions, and providing additional discussion. They have also included additional results showing that GluN2 photoantagonism* in vivo *in zebrafish induces axonal refinement deficits that are comparable to systemic MK801 exposure, providing another demonstration of utility. The authors have not provided data regarding the efficiency with which these photoswitches can be induced using two photon absorption, but they indicate that new MAG photoswitches will be developed based on strategies already shown to work with kainate and metabotropic receptors. It would have been nice to include data on the existing molecules to guide other investigators with their use, but I don't see this as an essential revision.*

In the text, we mention that 2-photon (2P) activation of LiGluN should be possible based on recent advances by our lab that demonstrated 2P photo-agonism in the kainite-type ionotropic GluR, iGluR6 (GluK2) (aka “LiGluR”), using one MAG variant, as well as in a metabotropic GluR, mGluR3 (aka “LimGluR3”), using another MAG variant (Carroll et al., 2015), as well as other MAG variants for LiGluR, based on a distinct photoswitch design; made by the Gorostiza lab (Izquierdo-Serra et al., 2014; Gascon-Moya et al., 2015). We have not yet generated MAG variants of these 2P types for LiGluNs.